# MAPEVAL: A MAP-BASED EVALUATION OF GEO-SPATIAL REASONING IN FOUNDATION MODELS

## ABSTRACT

Recent advancements in foundation models have enhanced AI systems' capabilities in autonomous tool usage and reasoning. However, their ability in location or map-based reasoning - which improves daily life by optimizing navigation, facilitating resource discovery, and streamlining logistics - has not been systematically studied. To bridge this gap, we introduce MAPEVAL, a benchmark designed to assess diverse and complex map-based user queries with geo-spatial reasoning. MAPEVAL features three task types (textual, API-based, and visual) that require collecting world information via map tools, processing heterogeneous geo-spatial contexts (e.g., named entities, travel distances, user reviews or ratings, images), and compositional reasoning, which all state-of-the-art foundation models find challenging. Comprising 700 unique multiple-choice questions about locations across 180 cities and 54 countries, MAPEVAL evaluates foundation models' ability to handle spatial relationships, map infographics, travel planning, and navigation challenges. Using MAPEVAL, we conducted a comprehensive evaluation of 28 prominent foundation models. While no single model excelled across all tasks, Claude-3.5-Sonnet, GPT-4o, and Gemini-1.5-Pro achieved competitive performance overall. However, substantial performance gaps emerged, particularly in MAPEVAL-API, where agents with Claude-3.5-Sonnet outperformed GPT-4o and Gemini-1.5-Pro by 16% and 21%, respectively, and the gaps became even more amplified when compared to open-source LLMs. Our detailed analyses provide insights into the strengths and weaknesses of current models, though all models still fall short of human performance by more than 20% on average, struggling with complex map images and rigorous geo-spatial reasoning. This gap highlights MAPEVAL's critical role in advancing general-purpose foundation models with stronger geo-spatial understanding.

## 1 INTRODUCTION

Recent advancements in foundation models, particularly large language models (LLMs) and vision-language models (VLMs), are significantly enhancing the capabilities of AI systems in autonomous tool usage (Qin et al., 2023; Yao et al., 2022) and reasoning (Lu et al., 2023; Wei et al., 2022). These developments facilitate the automation of everyday tasks through natural language instructions, especially in domains that require interaction with specialized tools like map services.

As platforms such as Google Maps or Apple Maps have become ubiquitous for accessing various location-based services (a.k.a tools/APIs) —ranging from finding nearby restaurants to determining the fastest routes between origins and destinations—there has been a growing interest in integrating maps with foundation models (Xie et al., 2024; Zheng et al., 2024). A couple of recent initiatives, such as WebArena (Zhou et al., 2023) and VisualWebArena (Koh et al., 2024), have introduced new tasks that involve map usage in practical scenarios.

However, despite the widespread adoption of map services and the promising potential of interactions between foundation models (e.g., LLMs and VLMs) and these services, no existing studies have rigorously tested the capabilities of foundation models in location or geo-spatial reasoning. This gap is critical, as effective map-based reasoning can optimize navigation, facilitate resource discovery, and streamline logistics in everyday life. Addressing this gap is essential for advancing the practical utility of AI in real-world applications.

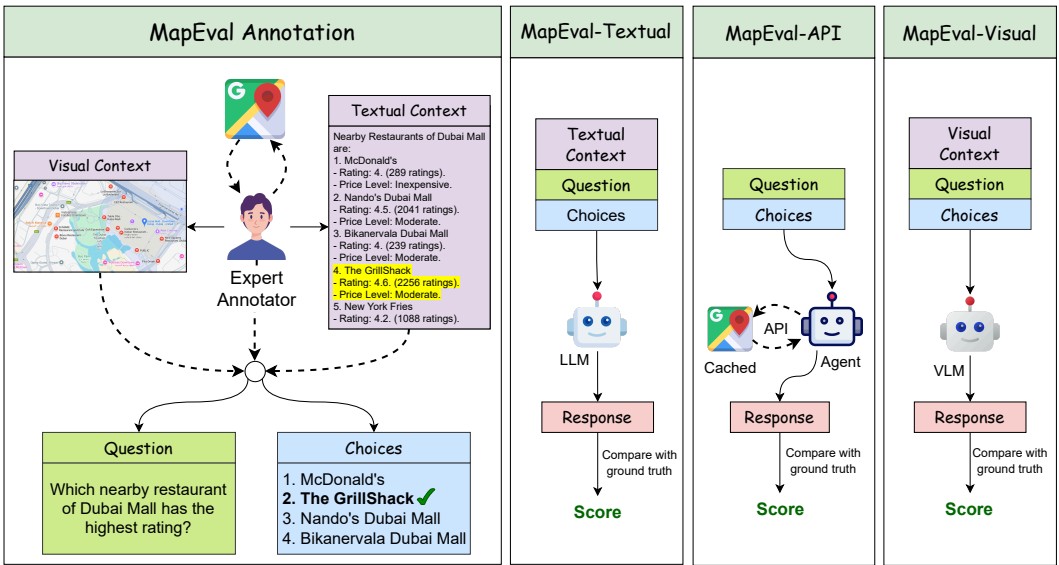

Figure 1: Overview of MAPEVAL. On the left, we show the annotation process, where an expert gathers either visual snapshots or textual data from Google Maps to create multiple-choice questions with ground truth labels. On the right, we depict the evaluation process and input/output for the three benchmark tasks in MAPEVAL.

We introduce MAPEVAL, a novel benchmark designed to evaluate the geo-spatial reasoning capabilities of foundation models and AI agents in complex map-based scenarios. MAPEVAL addresses a critical gap in existing benchmarks by evaluating models' ability to process heterogeneous geo-spatial contexts, perform compositional reasoning, and interact with real-world map tools. It features three task types— API, VISUAL, and TEXTUAL—that require models to collect world information via map tools, a deep visual understanding, and reason over diverse geo-spatial data (e.g., named entities, coordinates, operational hours, distances, routes, user reviews/ratings, map images), all of which remain challenging for state-of-the-art foundation models. Comprising 700 unique multiple-choice questions across 180 cities and 54 countries, MAPEVAL reflects real-world user interactions with map services while pushing state-of-the-art models to understand spatial relationships, map infographics, travel planning, POI search, and navigation. MAPEVAL ensures geographic diversity, realistic query patterns, and evaluation across multiple modalities. By integrating long contexts, visual complexity, API interactions, and questions requiring commonsense reasoning or recognition of insufficient information (i.e., unanswerability), it offers a rigorous framework for advancing geo-spatial AI capabilities. In Fig 1, we depict an overview of MAPEVAL.

With MAPEVAL, we evaluated 28 prominent foundation models, where Claude-3.5-Sonnet, GPT-4o, and Gemini-1.5-Pro showed competitive performance overall. However, significant gaps emerged in MAPEVAL-API, with Claude-3.5-Sonnet agents outperforming GPT-4o and Gemini-1.5-Pro by 16% and 21%, respectively, and even larger disparities compared to open-source models. Our detailed analyses revealed further insights into model strengths and weaknesses. Despite these advances, all models still fall short of human performance by over 20%, especially in handling complex map images and rigorous reasoning, underscoring MAPEVAL's role in advancing geo-spatial understanding. The benchmarking dataset and evaluation code will be open-sourced at https://github.com/MapEval.

## 2 RELATED WORK

Geo-spatial question answering presents significant challenges for foundation models (Mai et al., 2023). Early research in GeoQA (Mai et al., 2021) has focused on template-based methods (Zelle and Mooney, 1996; Chen et al., 2013; Chen, 2014; Punjani et al., 2018; Kefalidis et al., 2023), where predefined templates classify queries and retrieve information from structured databases like Open-StreetMap or DBpedia (Auer et al., 2007). While effective in certain scenarios, these methods are constrained by the static nature of the databases and the predefined templates, limiting their flex-

ibility in handling complex or dynamic queries. There has been limited effort to assess (Roberts et al., 2023) and improve (Balsebre et al., 2024) LLMs' capabilities in geospatial reasoning. Recent benchmarks such as Travel Planner (Xie et al., 2024), ToolBench (Qin et al., 2023), and API-Bank (Li et al., 2023) integrate map tools and APIs for location-based queries. While these benchmarks handle real-world tasks like itinerary planning or querying map data, the use of map APIs is limited to more straightforward use cases, such as calculating distances or identifying nearby points of interest. In addition, remote sensing research (Bastani et al., 2023; Yuan et al., 2024; Zhang et al., 2024; Lobry et al., 2020) has focused on extracting physical features from satellite imagery. While valuable for environmental monitoring and urban planning, this approach differs significantly from the task of reasoning over interactive digital map views, which involve understanding spatial relationships, map symbols, and navigation elements in a dynamic, user-interactive context.

## 3  THE MAPEVAL DATASET

### 3.1  DESIGN PRINCIPLES

**Reasoning.**  Geo-spatial reasoning in map-based tasks presents distinct challenges for foundation models, including: (a) understanding complex problem descriptions in natural language, (b) collecting relevant world information using map tools or APIs, (c) performing compositional and spatio-temporal reasoning, (d) interpreting map visuals, and (e) synthesizing information from heterogeneous geo-spatial contexts (e.g., named entities, distances, and temporal values). These tasks test the limits of state-of-the-art models, which struggle to fully grasp geo-spatial relationships, navigation complexities, and POIs.

**Realistic.**  MAPEVAL reflects real-world map usage by capturing typical user interactions with map services, such as: (a) varied usage patterns like location-based searches and travel planning, and (b) informal, often fragmented queries, without relying on perfect grammar or structure.

**Diversity.**  MAPEVAL ensures geographic diversity and broad evaluation across models and tasks: (a) capturing locations across cities and countries globally, and (b) offering a wide variety of question types and contexts, which test foundation models' spatial, temporal, data retrieval, and visual reasoning abilities.

**Long Contexts, Multi-modality, API Interactions.**  MAPEVAL challenges models with: (a) long geo-spatial descriptions, including POIs and navigational data, (b) complex map-specific images with location markers, and (c) API interactions, testing models' abilities as language agents in real-world map-based tasks.

**Unanswerability, Commonsense.**  MAPEVAL includes questions where context is insufficient to provide an answer, testing models' ability to identify missing or incomplete information, rather than making incorrect guesses. It also assesses commonsense reasoning and handling uncertainty, essential for reliable decision-making in real-world applications.

**Multiple Choice Questions (MCQs).**  We employ MCQs in MAPEVAL, similar to MMLU (Hendrycks et al., 2020), rather than open-ended queries. This approach circumvents the evaluation challenges associated with generated responses (Sai et al., 2022), allowing for a more straightforward and reliable accuracy-based assessment of map-based reasoning capabilities.

### 3.2  TASKS

**Textual.**  The objective of MAPEVAL-TEXTUAL is to answer MCQs by decomposing complex queries and extracting relevant information from long textual contexts. These contexts describe map locations, POIs, routes, navigation details, and travel distances/times, often including user ratings or reviews. Unlike typical reading comprehension tasks, these texts combine structured data (e.g., coordinates, distances) with unstructured narratives and subjective content. The model must reason over this heterogeneous information to select the correct answer. This task evaluates the model's ability to analyze fine-grained map-related information presented in text.

**API.**  In the MAPEVAL-API task, an AI agent interacts with map-based APIs to retrieve data (e.g., nearby POIs, distance calculations). The task involves generating API queries based on user questions, interpreting the returned structured data, and integrating it into reasoning processes to answer

| Type | Task | Question Example | Count |
|------|------|------------------|-------|
| Place Info | Textual/API | What is the direction of Victoria Falls from Harare? | 64 |
| | Visual | Is there any Hospital marked with a star symbol on the tourist map of Rome? | 121 |
| Nearby | Textual/API | Find restaurants nearby Louvre Museum above 4.0 rating. | 83 |
| | Visual | I stayed at SpringHill Suites by Marriott Portland Hillsboro. Can you recommend the nearest restaurant to my location? | 91 |
| Routing | Textual/API | I am driving to Brassica in Bexley Via E Whittier St. After reaching Lockbourne Rd, where should I go next? | 66 |
| | Visual | What is the fastest route from Times Square to Central Park by walking? | 80 |
| Unanswerable | Textual/API | Which road should I follow from Wola to Mokotów to avoid flooded roads in heavy rains? | 20 |
| | Visual | Which way should be efficient while visit from Abis bus station to KONO so that Victoria park is on the way | 20 |
| Trip | Textual/API | I have an afternoon free in New York and plan to visit The Metropolitan Museum of Art for 3 hours, followed by a 30-minute coffee break at a nearby cafe, and then spend 1 hour in Central Park. Plan a schedule to ensure I have enough time for everything. | 67 |
| Counting | Visual | How many hospitals are there in the left side of the river? | 88 |

Table 1: Examples of different question categories. MAPEVAL-TEXTUAL and MAPEVAL-VISUAL questions are accompanied by both textual and visual context (See appendix F for full qualitative example queries, contexts and evaluation model outputs during evaluations.)

**MCQs.** This task evaluates the model's ability to handle data retrieval, API interactions, and the synthesis of structured information in real-world, map-driven scenarios.

**Visual.** MAPEVAL-VISUAL task requires the model to interpret and analyze map snapshots, specifically digital map views from services like Google Maps. These snapshots represent complex spatial relationships, routes, landmarks, OCR texts (e.g., rating,) and symbolic elements (e.g., logos or traffic signs), which differ from typical image recognition tasks. The model must extract relevant information from the visuals, integrate it with spatial reasoning, and use it to answer MCQs. This task assesses the model's ability to tackle map-specific visual contents and perform spatial reasoning.

### 3.3 DATASET CONSTRUCTION

**Data Annotation.** To create a high-quality benchmark dataset for MAPEVAL, we utilized Google Maps, a widely adopted map service. The process of constructing the textual context presented significant challenges, particularly in ensuring accuracy and efficiency. For an example question like "What are the opening hours of the British Museum?" requires precise data to provide valid options and a correct answer. Manually searching for the "British Museum" on Google Maps and looking for its opening hours can be both time-consuming and prone to errors, making this method inefficient. To address these challenges, we employed MapQaTor, a web interface built on Google Maps APIs, designed to streamline the collection of textual map data. MapQaTor automates data retrieval from map APIs, collecting key information like opening hours and location details to build the textual dataset (Details in Appendix B.1). For each user query, we first fetch the necessary context data using MapQaTor. Questions were then paired with their corresponding contexts, and multiple-choice options were carefully curated based on this information. The ground truth answers were derived from the same context.

For MAPEVAL-API, the same questions were used as in MAPEVAL-TEXTUAL, but without textual contexts, requiring the language agents to interact with tools directly. To address consistency issues with real-time data updates, we created a controlled evaluation environment. This involves caching place information and simulating API interactions. Details of the pseudo-Google Maps setup are provided in Appendix C.

For the visual context, we capture map snapshots from Google Maps, covering random locations across various cities and countries worldwide. Based on each snapshot, we formulate relevant questions with multiple-choice options, where the correct labels are derived directly from the map information. To maintain traceability, we save the Google Maps URL for each snapshot. Additionally, to examine model capabilities at different zoom levels, we capture snapshots at varying zoom depths[1].

---

[1]Zoom levels found in map URLs indicate depth (e.g., url has zoom level 16.71), with higher values (e.g., 16 and above) showing more detail, compared to level 1 (world map)- See Appendix G.1

| Statistics | Number |
|---|---|
| Total unique question instances | 700 |
| - Questions with api or textual-context | 300 (42.86%) |
| - Questions with visual-context | 400 (57.14%) |
| Total unique countries | 54 |
| Total unique cities | 180 |
| Maximum textual-context length | 1500 |
| Maximum question length | 107 |
| Maximum questions from a country | 132 |
| Maximum questions from a city | 44 |
| Average textual-context length | 435.63 |
| Average question length | 21.41 |
| Unique number of textual-context | 215 |
| Unique number of visual-context | 270 |
| Min, Max, Avg Choices | 2, 7, 4.004 |
| Max zoom of visual-context | 21.0 |
| Min zoom of visual-context | 8.0 |
| Average zoom of visual-context | 15.26 |

Table 2: Key statistics of MAPEVAL. Lengths are in words. Visual-context means Map snapshots/images. Some questions are yes/no and some have additional complexity with 4+ choices.

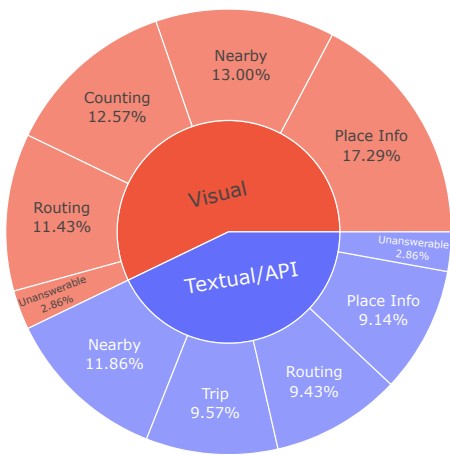

Figure 2: MAPEVAL category statistics.

We create the following question types for MAPEVAL: (a) Place Info: detects POIs and asks about specific details related to a place (e.g., location, rating, reviews); (b) Nearby: identifies nearby places or POIs; (c) Routing: navigates between locations, considering routes and landmarks; (d) Unanswerable: when the map information (e.g., from google map) or the textual and visual context is insufficient to answer the question. Note that, in each category we formulate a few questions that requires general knowledge or reasoning about locations and navigation (e.g., there are 52 commonsense QAs in MAPEVAL-VISUAL).

Moreover, MAPEVAL-TEXTUAL and MAPEVAL-API exclusively feature Trip questions, which involve planning multi-stop journeys across various POIs. Due to the complexity and details of trip planning, these questions are difficult to represent in a single visual snapshot. Conversely, Counting tasks are unique to MAPEVAL-VISUAL, where models count specific items or locations on a map—a challenge specifically tailored to visual contexts.

**Quality Control and Human Performance** To ensure quality, each QA pair is annotated by multiple members of our team, achieving an initial 76% mutual agreement. At least two team members then manually verify and resolve any disputes on the remaining pairs; if consensus cannot be reached (i.e., ambiguous), that pair is filtered out. To compute human scores, two team members who did not participate in the annotation process attempt to answer the questions, and their highest-scoring attempts are reported as the human performance benchmark. For MAPEVAL-API, as the questions are identical to MAPEVAL-TEXTUAL, we report the same human performance for both.

## 3.4 DATASET STATISTICS AND ANALYSIS

The main statistics of MAPEVAL are presented in Table 2 and Figure 2. Examples of each question type and their numbers are presented in Table 12. We visualize the global distribution of locations in our dataset using coordinates (Fig. 3). Table 13 (Appendix) lists all countries and their frequencies in MAPEVAL. We use OpenStreetMap's Nominatim API for reverse geocoding to determine countries from coordinates. Textual context includes the coordinates of places in it. In case of visual context, we can find the coordinates from the associated Map URL with each snapshot. For example, coordinate of an example url, is 35.7048455,139.763263. We visualize the distribution of question and textual context lengths in the Appendix (Figures 7 and 8). Overall, beyond their diversity in types, questions and contexts also vary significantly in length, reflecting varying levels of complexity and detail. Furthermore, in Appendix G.1, we illustrate the zoom level distribution in MAPEVAL-VISUAL, adding another dimension to the dataset's diversity and evaluation challenges.

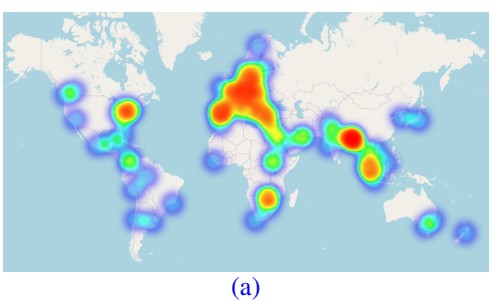 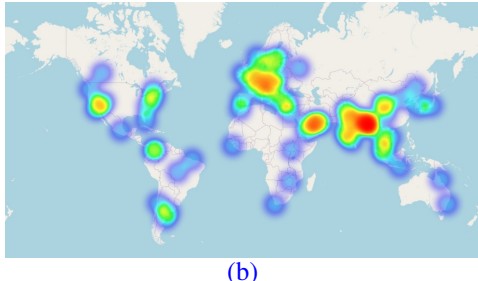

(a)                                                            (b)

Figure 3: Geographical Distribution of Textual and Visual Contexts. The left heatmap (a) represents the locations of places mentioned in textual contexts, while the right heatmap (b) shows the locations derived from map snapshots in visual contexts.

## 4 EXPERIMENTS

### 4.1 EXPERIMENTAL PROTOCOL AND SETUP

We evaluate all tasks using the accuracy metric, defined as the percentage of correct choices selected by the model. We prompt models with the respective context, question, tool usage documentations (only for MAPEVAL-API), answer format guidelines, and choices. We assess LLMs for MAPEVAL-TEXTUAL, VLMs for MAPEVAL-VISUAL, and ReACT agents Yao et al. (2022) (known for effective tool interaction (Zhuang et al., 2023)) built on various LLMs for MAPEVAL-API, aligning each task with appropriate model types. Appendix F presents example prompts for all tasks. Our LLMs and VLMs spans both open and closed-source models. Closed-source models include Claude-3.5-Sonnet, GPT-4o, GPT-4-Turbo (Achiam et al., 2023), GPT-3.5-Turbo (OpenAI, 2022), Gemini-1.5 (Pro, Flash; Reid et al. (2024)), with all except GPT-3.5-Turbo being multi-modal foundation models used in all tasks, while GPT-3.5-Turbo, which is text-only, is utilized solely in the MAPEVAL-TEXTUAL and MAPEVAL-API tasks. Open-source LLMs include instruct versions of Gemma-2.0 (9B, 27B; Team et al. (2024)), Llama-3.2 (3B, 90B), Llama-3.1 (8B, 70B; Vavekanand and Sam (2024)) , Mistral-Nemo-7B, Mixtral-8x7B (Jiang et al., 2024), Qwen2.5 (7B, 14B, 72B; Team (2024)) , Phi-3.5-mini. For MAPEVAL-VISUAL, we considered the open-source VLMs: Qwen2-VL-7B-Instruct (Wang et al., 2024), MiniCPM-Llama3-V-2_5 (Yao et al., 2024), Llama-3-VILA1.5-8B (Lin et al., 2023), glm-4v-9b (GLM et al., 2024), InternLm-xcomposer2 (Dong et al., 2024), paligemma-3b-mix-224 (Beyer et al., 2024), DocOwl1.5 (Hu et al., 2024), llava-v1.6-mistral-7b-hf (Liu et al., 2024b), and llava-1.5-7b-hf (Liu et al., 2024a). In MAPEVAL-API task, we concentrate our exploration on high-capacity open-source LLMs, specifically Llama-3.2-90B, Llama-3.1-70B, Mixtral-8x7B, and Gemma-2.0-9B. We limit our evaluation of open-source models in AI agents due to the task's complexity and resource demands, the lower performance of smaller models, and the excessive number of calls for both LLMs and map APIs.

### 4.2 RESULTS AND ANALYSIS

#### 4.2.1 MAPEVAL-TEXTUAL

We present MAPEVAL-TEXTUAL results summaries in Table 3. Our benchmarking reveals significant insights into the current state of geo-spatial reasoning capabilities in language models. The results demonstrate a clear performance hierarchy, with closed-source models generally outperforming their open-source counterparts. Claude-3.5-Sonnet leads with 66.33% overall accuracy, while the best open-source model, Llama-3.1-70B, achieves 61.00%. However, the substantial gap between even the top-performing models and human accuracy (86.67%) underscores the challenges that remain in geo-spatial reasoning tasks. Models generally excel in "Place Info", "Nearby", and "Routing" tasks (best performance ∼75%), benefiting from the comprehensive context extracted by MAPEVAL-TEXTUAL. This includes textual descriptions, opening hours, distances, and routing times, enabling LLMs to easily extract relevant information and perform basic mathematical reasoning. In contrast, models struggle significantly with "Trip" planning scenarios (best performance ∼49%), indicating difficulties with complex, multi-step reasoning. This poor performance is primarily due to the challenge of aggregating multiple routes with various spatio-temporal constraints, a task that remains universally difficult across model types. Performance on "Unanswerable" queries

| Model | Overall | Place Info | Nearby | Routing | Trip | Unanswerable |
|-------|---------|------------|--------|---------|------|--------------|
| Close-Source (Proprietary) LLMs | | | | | | |
| Claude-3.5-Sonnet | **66.33** | **73.44** | 73.49 | **75.76** | **49.25** | 40.00 |
| Gemini-1.5-Pro | **66.33** | 65.63 | **74.70** | 69.70 | 47.76 | **85.00** |
| GPT-4o | 63.33 | 64.06 | **74.70** | 69.70 | **49.25** | 40.00 |
| GPT-4-Turbo | 62.33 | 67.19 | 71.08 | 71.21 | 47.76 | 30.00 |
| Gemini-1.5-Flash | 58.67 | 62.50 | 67.47 | 66.67 | 38.81 | 50.00 |
| GPT-4o-mini | 51.00 | 46.88 | 63.86 | 57.58 | 40.30 | 25.00 |
| GPT-3.5-Turbo | 37.67 | 26.56 | 53.01 | 48.48 | 28.36 | 5.00 |
| Open-Source LLMs | | | | | | |
| Llama-3.1-70B | **61.00** | **70.31** | 67.47 | **69.70** | 40.30 | **45.00** |
| Llama-3.2-90B | 58.33 | 68.75 | 66.27 | 66.67 | 38.81 | 30.00 |
| Qwen2.5-72B | 57.00 | 62.50 | **71.08** | 63.64 | **41.79** | 10.00 |
| Qwen2.5-14B | 53.67 | 57.81 | **71.08** | 59.09 | 32.84 | 20.00 |
| Gemma-2.0-27B | 49.00 | 39.06 | **71.08** | 59.09 | 31.34 | 15.00 |
| Gemma-2.0-9B | 47.33 | 50.00 | 50.60 | 59.09 | 34.33 | 30.00 |
| Llama-3.1-8B | 44.00 | 53.13 | 57.83 | 45.45 | 23.88 | 20.00 |
| Qwen2.5-7B | 43.33 | 48.44 | 49.40 | 42.42 | 38.81 | 20.00 |
| Mistral-Nemo | 43.33 | 46.88 | 50.60 | 50.00 | 32.84 | 15.00 |
| Mixtral-8x7B | 43.00 | 53.13 | 54.22 | 45.45 | 26.87 | 10.00 |
| Phi-3.5-mini | 37.00 | 40.63 | 48.19 | 46.97 | 20.90 | 0.00 |
| Llama-3.2-3B | 33.00 | 31.25 | 49.40 | 31.82 | 25.37 | 0.00 |
| Human Performance | | | | | | |
| Human | 86.67 | 92.19 | 90.36 | 81.81 | 88.06 | 65.00 |

Table 3: MAPEVAL-TEXTUAL performances. Figure 12 visualizes the categorical accuracy.

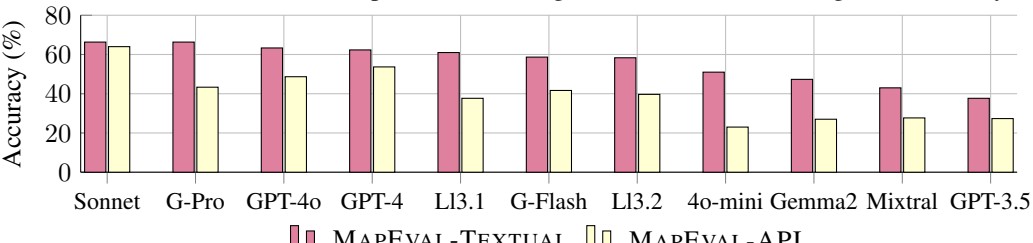

Figure 4: Comparison between MAPEVAL-TEXTUAL and MAPEVAL-API .

varies significantly, highlighting the importance of recognizing insufficient information in real-world applications. Gemini models, particularly Gemini-1.5-Pro with 85% accuracy, outperformed others in this category, where most models achieved only 0-45% accuracy. This stark contrast, along with consistent underperformance in "Trip" planning across all models, points to fundamental challenges in geo-spatial reasoning that transcend individual architectures. These findings validate our benchmark's ability to identify key areas for improvement in AI systems handling location-based queries and planning. Furthermore, the benchmark's results illustrate the impact of model scale, with larger models generally outperforming smaller ones. However, the performance gap between open and closed-source models suggests significant potential for advancements in open-source development, as Fig 13 highlights additional challenges for open-source models in handling longer contexts.

### 4.2.2  MAPEVAL-API

We present the MAPEVAL-API results in Table 4, highlighting key insights into the geo-spatial reasoning abilities of language models when interacting with map APIs. The analysis shows that MAPEVAL-API generally underperforms compared to MAPEVAL-TEXTUAL across most task categories, with significant performance drops observed in Nearby tasks (from 74.70% to 55.42%) and Routing tasks (from 75.76% to 65.15%). Figure 4 visualizes these differences across models. While Claude-3.5-Sonnet demonstrated consistent performance, other models experienced noticeable declines, primarily due to the absence of direct context and the complexity of tool usage. This highlights the need for a more advanced agent surpassing ReAct's capabilities in geo-spatial domains. Interestingly, in the Trip category, MAPEVAL-API achieved a notable improvement of approximately 22% in its best performance compared to MAPEVAL-TEXTUAL. This suggests that MAPEVAL-API is particularly effective at step-by-step reasoning required for solving multi-step complex problems. Claude-3.5-Sonnet led the results with an overall accuracy of 64.00%, show-

| Model | Overall | Place Info | Nearby | Routing | Trip | Unanswerable |
|---|---|---|---|---|---|---|
| Close-Source (Proprietary) LLMs | | | | | | |
| Claude-3.5-Sonnet | **64.00** | **68.75** | **55.42** | **65.15** | **71.64** | 55.00 |
| GPT-4-Turbo | 53.67 | 62.50 | 50.60 | 60.61 | 50.75 | 25.00 |
| GPT-4o | 48.67 | 59.38 | 40.96 | 50.00 | 56.72 | 15.00 |
| Gemini-1.5-Pro | 43.33 | 65.63 | 30.12 | 40.91 | 34.33 | **65.00** |
| Gemini-1.5-Flash | 41.67 | 51.56 | 38.55 | 46.97 | 34.33 | 30.00 |
| GPT-3.5-Turbo | 27.33 | 39.06 | 22.89 | 33.33 | 19.40 | 15.00 |
| GPT-4o-mini | 23.00 | 28.13 | 14.46 | 13.64 | 43.28 | 5.00 |
| Open-Source LLMs | | | | | | |
| Llama-3.2-90B | **39.67** | **54.69** | **37.35** | 39.39 | 35.82 | 15.00 |
| Llama-3.1-70B | 37.67 | 53.13 | 32.53 | **42.42** | 31.34 | 15.00 |
| Mixtral-8x7B | 27.67 | 32.81 | 18.07 | 27.27 | **38.81** | 15.00 |
| Gemma-2.0-9B | 27.00 | 35.94 | 14.46 | 28.79 | 26.87 | **45.00** |
| Human Reference | | | | | | |
| Human (MAPEVAL-TEXTUAL) | 86.67 | 92.19 | 90.36 | 81.81 | 88.06 | 65.00 |

Table 4: MAPEVAL-API evaluation performance (See Figure 14 to visualize categorical accuracy)

casing robust performance both as a tool agent and in generic graph reasoning beyond maps. A substantial performance gap remains between closed-source and open-source models, with the best open-source model, Llama-3.2 90B, achieving only 39.67% overall accuracy. Similar to MAPEVAL-TEXTUAL, performance on "Unanswerable" queries showed wide variation (5% to 65%), underscoring the need for models to better identify insufficient information in real-world scenarios.

### 4.2.3 MAPEVAL-VISUAL

We evaluate models on the MAPEVAL-VISUAL task in Table 5. As observed, closed-source models generally outperform their open-source counterparts, with Claude-3.5-Sonnet leading with an overall accuracy of 61.65%, followed by GPT-4o at 58.90% and Gemini-1.5-Pro at 56.14%. Among open-source models, Qwen2-VL-7B-Instruct tops the list with 51.63% overall accuracy. While the models perform well in Place Info tasks, achieving a high accuracy of 82.64%, they struggle with more complex tasks like Counting, Nearby, and Routing, indicating areas where current models require significant improvement. However, it is crucial to understand why models with strong image reasoning capabilities still underperform on map-specific tasks. We conjecture that they are well-trained on generic images but not on detailed map data. To validate, Fig 5 plots accuracy against zoom levels, showing a significant performance drop at higher zoom depths (e.g., streets, symbols, demarcations) beyond level 14, where map details become more complex. Our benchmark dataset exposes a substantial performance gap between AI models and human performance, particularly in tasks that require nuanced reasoning. For instance, human performance on Routing tasks (85.18%) far surpasses the best model's accuracy (50%), and a similar gap is seen in the Counting task (78.41% for humans versus 47.73% for the best AI). Additionally, the dataset highlights disparities in handling uncertainty: while models like Claude-3.5-Sonnet and Gemini-1.5-Pro excel in identifying unanswerable questions, with accuracy rates of 90% and 80%, other models, especially open-source ones, struggle significantly.

### 4.3 QUALITATIVE ERROR ANALYSIS

LLMs face challenges in spatial, temporal, and commonsense reasoning when answering location-based queries. In spatial reasoning, they struggle with straight-line distances (Example at Listing 1), cardinal directions (e.g., East, West, North, South; Example at Listing 2), and step-by-step route planning, leading to decreased accuracy, particularly with math or counting (e.g., nearby restaurant counts; Example at Listing 3). Temporal reasoning issues include failing to plan trips efficiently or calculate optimal visiting times, such as errors in travel times or visit durations (Example at Listing 4). Commonsense reasoning failures occur when models cannot deduce simple conclusions from context and often hallucinates (Example at Listing 5). LLM-based agents also face challenges using map tools or APIs, particularly in Nearby and Routing queries. Misuse or misinterpretation of parameters leads to failed results, such as omitting key parameters or using incompatible values. When encountering no valid routes or results, agents may fall into infinite loops, repeatedly issuing identical requests without adjusting their approach. These issues highlight the need for better API handling and error recovery mechanisms. In visual tasks, VLMs often struggle with spatial awareness, showing confusion when POIs are visually close together or incorrectly identifying and

| Model | Overall | Place info | Nearby | Routing | Counting | Unanswerable |
|---|---|---|---|---|---|---|
| Claude-3-5-Sonnet | **61.65** | **82.64** | 55.56 | 45.00 | **47.73** | **90.00** |
| GPT-4o | 58.90 | 76.86 | **57.78** | **50.00** | **47.73** | 40.00 |
| Gemini-1.5-Pro | 56.14 | 76.86 | 56.67 | 43.75 | 32.95 | 80.00 |
| GPT-4-Turbo | 55.89 | 75.21 | 56.67 | 42.50 | 44.32 | 40.00 |
| Gemini-1.5-Flash | 51.94 | 70.25 | 56.47 | 38.36 | 32.95 | 55.00 |
| GPT-4o-mini | 50.13 | 77.69 | 47.78 | 41.25 | 28.41 | 25.00 |
| Open-Source VLMs | | | | | | |
| Qwen2-VL-7B-Instruct | **51.63** | 71.07 | **48.89** | 40.00 | **40.91** | **40.00** |
| Glm-4v-9b | 48.12 | **73.55** | 42.22 | 41.25 | 34.09 | 10.00 |
| InternLm-Xcomposer2 | 43.11 | 50.41 | **48.89** | **43.75** | 34.09 | 10.00 |
| MiniCPM-Llama3-V-2_5 | 40.60 | 60.33 | 32.22 | 32.50 | 31.82 | 30.00 |
| Llama-3-VILA1.5-8B | 32.99 | 46.90 | 32.22 | 28.75 | 26.14 | 5.00 |
| DocOwl1.5 | 31.08 | 43.80 | 23.33 | 32.50 | 27.27 | 0.00 |
| Llava-v1.6-Mistral-7B-hf | 31.33 | 42.15 | 28.89 | 32.50 | 21.59 | 15.00 |
| Paligemma-3B-mix-224 | 30.58 | 37.19 | 25.56 | 38.75 | 23.86 | 10.00 |
| Llava-1.5-7B-hf | 20.05 | 22.31 | 18.89 | 13.75 | 28.41 | 0.00 |
| Human Performance | | | | | | |
| Human | 82.23 | 81.67 | 82.42 | 85.18 | 78.41 | 65.00 |

Table 5: MAPEVAL-VISUAL evaluation performance. (Fig 16 visualizes categorical accuracy).

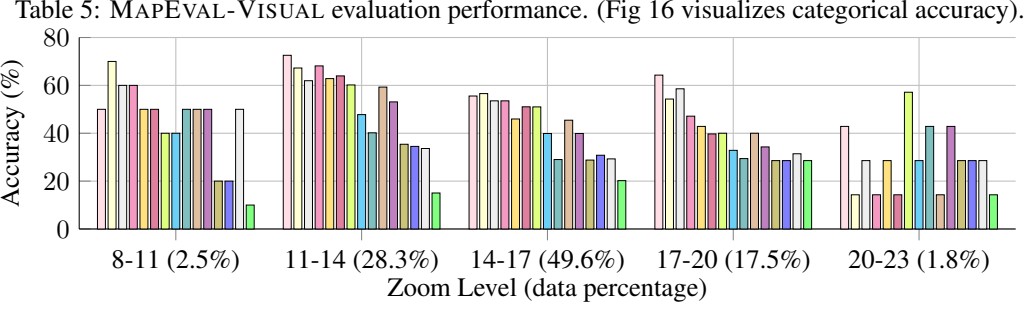

Figure 5: Accuracy by Zoom Level.

counting POIs in map images (e.g., malls/stores). Such errors underscore the need for enhanced spatial awareness, temporal reasoning, and tool usage in foundation models (details in Appendix E).

# 5 ENHANCING GEOSPATIAL REASONING IN FOUNDATION MODELS

**Calculator Integration for Complex Spatial Computations:** In MAPEVAL-TEXTUAL, LLMs showed significant variability in their ability to perform spatial reasoning tasks like calculating straight-line distances (see Fig. 17), cardinal directions (see Fig. 18) and counting-related queries (see Fig. 19). For example: (i) Claude-3.5-Sonnet achieved the highest accuracy (91%) in identifying cardinal directions, while Gemma-2.0-27B scored the lowest (16.67%). (ii) In measuring straight-line distances, all models struggled, with the best accuracy being only 51.06%. (iii) Counting tasks posed challenges, even for dominating models like Claude-3.5-Sonnet, which underperformed compared to the open-source Gemma-2.0-27B (60.87% accuracy). To address these issues, we extended model capabilities by providing access to external tools (e.g., calculator) specifically for calculating straight-line distances and cardinal directions (Details in Appendix I). This resulted in a dramatic improvement (see Table 6), with accuracies increasing by over 50% in certain cases. For instance: (i) The accuracy of Claude-3.5-Sonnet in calculating straight-line distances increased from 51.06% to 85.11%, demonstrating the utility of integrating external tools. (ii) GPT-4o-mini, which initially struggled with cardinal direction tasks, saw its performance increase from 29.17% to 91.67%, showcasing a remarkable transformation with tool support. (iii) Even open-source models like Gemma-2.0-9B benefited, achieving an accuracy boost in straight-line distance tasks from 29.79% to 68.90%. These improvements highlight the challenges LLMs face when reasoning spatially without external support, especially in complex or unfamiliar contexts. By leveraging tools, models can offload computationally intensive or context-specific reasoning tasks, enabling more

| Model | Straight-Line Distance | | Cardinal Direction | |
|---|---|---|---|---|
| | LLM | LLM+Calculator | LLM | LLM+Calculator |
| Claude-3.5-Sonnet | 51.06 | 85.11 | 91.67 | 95.83 |
| GPT-4o | 46.81 | 70.21 | 62.50 | 87.50 |
| GPT-4-Turbo | 40.43 | 76.59 | 58.33 | 91.67 |
| Gemini-1.5-Pro | 38.29 | 72.34 | 62.50 | 91.67 |
| Gemini-1.5-Flash | 46.81 | 63.83 | 58.33 | 87.50 |
| GPT-4o-mini | 34.04 | 78.72 | 29.17 | 91.67 |
| GPT-3.5-Turbo | 19.15 | 55.32 | 20.83 | 62.50 |
| Llama-3.2-90B | 42.55 | 68.90 | 66.67 | 87.50 |
| Llama-3.1-70B | 48.94 | 61.7 | 66.67 | 95.83 |
| Mixtral-8x7B | 38.29 | 59.57 | 33.33 | 79.17 |
| Gemma-2.0-9B | 29.79 | 68.09 | 37.50 | 75.00 |

Table 6: Performance Improvement of LLMs in Straight-Line Distance and Cardinal Directions Analysis (Fig. 20 and 21 visualizes the improvement).

precise and reliable results. However, spatial reasoning is only one aspect of location-based tasks where models continue to underperform. For instance, temporal reasoning tasks, such as incorporating travel times and determining optimal visiting hours, could benefit from additional tools. Expanding tool integration in this way could improve the model's performance across multiple reasoning domains, but it would also add significant complexity to the architecture, requiring the management of multiple tools for different types of reasoning.

**Adaptive Routing of Tools and Models:** In ReAct-based systems, a significant challenge arises from the heavy responsibility placed on a single agent to extract relevant parameters from a question, call APIs with those parameters, and then provide the final answer based on API responses. This complex process often leads to issues such as parameter extraction errors, incorrect API calls, or dead loops (e.g., GPT-3.5-Turbo encountering 16 infinite iterations; see Fig. 15). These problems are particularly evident when the agent is unable to effectively reason through the task, reducing task completion rates. In fact the processing of large amount of API data even in plain text form (i.e., long contexts in MAPEVAL-TEXTUAL task) pose a significant challenge to LLMs (i.e., as discussed in Section 4.2.1 as well as the low performances in Table 3). To address these limitations, the CHAMELEON Framework (Lu et al., 2024) offers a robust solution that adaptively breaks the task into multiple tool usage modules (e.g., multi-agent system). The integration of CHAMELEON into the MAPEVAL-API has already shown a notable improvement in GPT-3.5-Turbo's performance, (Table 7). Besides, CHAMELEON's ability to decompose tasks and handle errors more efficiently results in fewer parameter extraction errors and prevents dead loops, significantly boosting accuracy. Another promising alternative approach would be to develop an ensemble system that combines a query classifier with type-specific LLM deployment. This system would first classify incoming queries and then route them to the best-performing LLM for that particular query type achieving potential superiority.

| Category | ReAct | CHAMELEON |
|---|---|---|
| Place Info | 39.06 | 54.69 |
| Nearby | 22.89 | 54.21 |
| Routing | 33.33 | 51.51 |
| Trip | 19.40 | 43.28 |
| Unans. | 15.00 | 25.00 |
| Overall | 27.33 | 49.33 |

Table 7: Accuracy with GPT-3.5-Turbo

# 6  CONCLUSION

In this paper, we introduce MAPEVAL, a comprehensive benchmark dataset designed to assess foundation models in geo-spatial reasoning through *textual*, *API-based*, and *visual* evaluation modes. MAPEVAL incorporates diverse real-world scenarios to thoroughly evaluate model capabilities on geo-spatial reasoning tasks. Our findings reveal that while leading models like Claude-3.5-Sonnet, GPT-4o, and Gemini-1.5-Pro excel in certain areas, they still significantly underperform compared to human accuracy, especially when using open-source foundation models. This highlights critical areas for improvement, especially in managing complex map-based queries that require multi-step spatio-temporal reasoning, efficient tool utilization, and domain-specific knowledge. Future work could focus on developing specialized geospatial models, integrating LLMs with external tools like map APIs, and enhancing VLMs' visual understanding of map images. We anticipate that MAPEVAL will catalyze ongoing research in geospatial reasoning and broader QA domains.

## REPRODUCIBILITY STATEMENT

To ensure the reproducibility of our results, we provide the evaluation codes and the complete dataset used for our experiments at: `https://github.com/MapEval`. The inference process for the LLMs, including parameters such as temperature, top-k, and top-p, is part of the evaluation code. Any updates or bug fixes will be made available in the repository. The organization is anonymous due to the double-blind review process.

## LIMITATIONS

Our dataset does not cover all available Google Maps APIs, which limits the scope of our evaluation. Specifically, we have used five APIs from the Places and Routes categories: Text Search, Place Details, Nearby Search, Directions, and Distance Matrix. However, we did not incorporate other API categories such as Maps and Environment. This restricted API usage narrows the variety of queries we could evaluate and may leave out other valuable geospatial insights that could be gained from broader API usage.

Furthermore, any future updates to the APIs we used may not be reflected in our dataset, which could impact its relevance for real-time applications, potentially making it outdated and more suitable for archival purposes.

Another limitation is that the performance observed in our evaluation may not transfer to other domains or tools, as we did not explore this possibility. The generalizability of our methods remains an area for future research.

Finally, different prompt formulations could lead to variations in the results, but we did not experiment with this aspect. Future work could focus on examining how different prompts affect the LLM's performance in geospatial reasoning tasks.

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

## A    DETAILED RELATED WORK

### A.1    MAPEVAL-TEXTUAL

Template-based GeoQA models (Zelle and Mooney, 1996; Chen et al., 2013; Chen, 2014; Punjani et al., 2018; Kefalidis et al., 2023) have predominantly followed a two-step strategy for answering geographic questions: (1) classifying a natural language query into predefined templates and (2) using these templates to query structured geographic knowledge sources such as PostGIS, DBpedia (Auer et al., 2007), YAGO (Suchanek et al., 2007), Freebase (Bollacker et al., 2007), and OpenStreetMap. While these approaches are effective for structured queries, they are limited by the predefined question templates and their reliance on static databases. They typically convert natural language questions into structured query language scripts. For instance, GeoQuestions1089 (Kefalidis et al., 2023) contains 1089 questions with corresponding GeoSPARQL (Open Geospatial Consortium, 2011) queries over the YAGO2geo (Karalis et al., 2019) geospatial knowledge graph.

In contrast, our MAPEVAL-TEXTUAL approach shifts the focus from database querying to assessing geospatial reasoning in Large Language Models (LLMs). Annotators collect factual map services data using MapQaTor, which is then provided as context to LLMs. This setup isolates and evaluates the model's ability to reason over geospatial relationships, addressing the challenge of free-form, complex map-related queries in a dynamic environment. This approach allows for a more holistic evaluation of LLMs, reflecting real-world usage where users interact with map tools using natural language queries. Thus, in MapEval, the responsibility lies with LLMs to answer the questions, whereas in previous works, the models were tasked with generating queries (e.g., Geoquery, GeoSPARQL), which are used to query external knowledge bases.

GPT4GEO (Roberts et al., 2023) explored GPT-4's factual geographic knowledge by characterizing what it "knows" about the world without plugins or Internet access. Their evaluation focused on analyzing a single model using templated queries about generic location and direction-oriented facts, such as routing, navigation, and planning for well-known cities and places. However, this approach is inherently constrained by the training data of GPT-4, making it incapable of answering questions about less-known places. While the findings suggest that GPT-4 shows promising geo-spatial knowledge, this approach neither establishes a benchmark for geo-spatial reasoning nor incorporates real-life user queries or map services (e.g., Google Maps) as a geospatial information base.

Our approach employs fundamentally different evaluation and design principles. We establish a benchmarking of deeper geo-spatio-temporal reasoning capabilities across multiple foundation models using real user queries rather than templates. Uniquely, our evaluation encompasses multimodal understanding, tool interactions, and answerability determination. Additionally, we provide foundation models with fine-grained map services data through both context and API access, enabling a more comprehensive benchmarking of their geospatial question-answering abilities.

### A.2    MAPEVAL-API

The MAPEVAL-API task adopts a practical approach by leveraging map APIs to answer location-based questions directly, providing a more real-world scenario for evaluating the capabilities of Large Language Models (LLMs) in map-based reasoning. Recent advancements in LLMs have led to growing interest in planning tasks (Xie et al., 2024; Balsebre et al., 2024; Zheng et al., 2024; Fang et al., 2024) that involve map data. For instance, the Travel Planner (Xie et al., 2024) benchmark assessed multi-day itinerary planning using Google Maps API to determine distances, travel times, and details of nearby attractions. This task demonstrated the utility of map data in real-world planning scenarios, highlighting the potential for LLMs to integrate real-time geospatial information into decision-making.

Additionally, tool-calling benchmarks such as ToolBench (Qin et al., 2023) and API-Bank (Li et al., 2023) have included location-based queries as a subtask, testing the ability of LLMs to interact with APIs in structured ways. These benchmarks typically focus on simpler query types, such as retrieving distances or nearby points of interest (POIs), but they do not fully address the complexity and diversity of real-world map-based questions.

In contrast, MapEval-API pushes the boundaries by evaluating LLMs on a wide variety of complex geospatial tasks that require not only querying map APIs but also integrating multiple pieces of information, such as travel itineraries, nearby services, and spatio-temporal reasoning. This more comprehensive evaluation of API-based reasoning challenges the models to process complex, multi-faceted questions, highlighting their ability to handle nuanced map interactions and effectively synthesize data retrieved from APIs.

### A.3 MAPEVAL-VISUAL

Prior works in geospatial analysis and map-based question answering have predominantly focused on remote sensing images (Bastani et al., 2023; Yuan et al., 2024; Zhang et al., 2024), which involve satellite or aerial imagery. These images often contain complex data about the Earth's surface, including land cover, vegetation, urban infrastructure, and other environmental features. Models designed for interpreting remote sensing images (Lobry et al., 2020) typically rely on convolutional neural networks (CNNs) and other computer vision techniques for object detection, segmentation, and classification tasks. These methods often focus on identifying physical entities like roads, buildings, and natural features from high-resolution imagery.

In contrast, our MAPEVAL-VISUAL approach focuses on digital map view snapshots, which are 2D representations of map services (such as Google Maps). Unlike remote sensing images, which represent physical realities captured from a top-down perspective, these digital maps show geospatial information in a structured, interactive format. The focus of MAPEVAL-VISUAL is to evaluate a model's ability to interpret and reason about these structured map views, which include not just physical features, but also symbolic and navigational information such as traffic signs, routes, landmarks, and visual cues from the map interface itself.

While remote sensing image analysis typically involves extracting physical data from raw image pixels, MAPEVAL-VISUAL requires models to engage with spatial reasoning and map-based symbols, demanding a different set of computational skills. In this task, the model must not only understand the spatial relationships between map features but also reason about the context provided by digital map interfaces, which include additional elements such as zoom levels, icons, and navigation markers. This distinction sets MAPEVAL-VISUAL apart from traditional remote sensing tasks and presents new challenges in the field of geospatial reasoning and map-based visual question answering.

## B DATA COLLECTION DETAILS

### B.1 MAPQATOR: ANNOTATOR INTERFACE

For the creation of the textual contexts and design MCQs based on that, we employed a custom-built web interface named **MapQaTor**. As illustrated in Figure 6, this interface was central to the dataset development process, offering an intuitive, user-friendly environment that simplifies complex tasks, such as API interaction and context generation.

The annotator interface is designed to reduce technical complexity for users, allowing them to concentrate on the core aspects of dataset annotation, such as selecting relevant locations, providing information on distances, durations, and directions between places, as well as identifying nearby points of interest. Its streamlined workflow facilitates efficient dataset creation by automating repetitive tasks, which not only minimizes errors but also significantly accelerates the annotation process.

MapQaTor uses five key Google Maps APIs: Text Search, Place Details, Distance Matrix, Directions, and Nearby Search, based on their relevance to common map-based tasks and their ability to provide comprehensive location data.

MapQaTor caches all API call responses, creating a static database for evaluation purposes. This ensures consistent responses when evaluating MAPEVAL-API. Specifically, when an API call is made, the cached response is returned instead of a real-time query, maintaining a controlled and static evaluation environment.

Once the dataset is generated, it can be easily exported in JSON format, making it readily usable for further analysis and evaluation in downstream tasks, such as model training and benchmarking.

### B.2 FILTERING VIA LLMS:

To ensure the challenge and quality of our dataset, we evaluated a range of LLMs. We filtered out samples where the majority of the LLMs could easily provide the correct answer, considering these samples "too easy" and removing them from the dataset. Additionally, we identified samples where most LLMs failed to answer the questions based on the given context. In such cases, we re-examined the questions, correcting any inconsistencies to improve clarity and relevance.

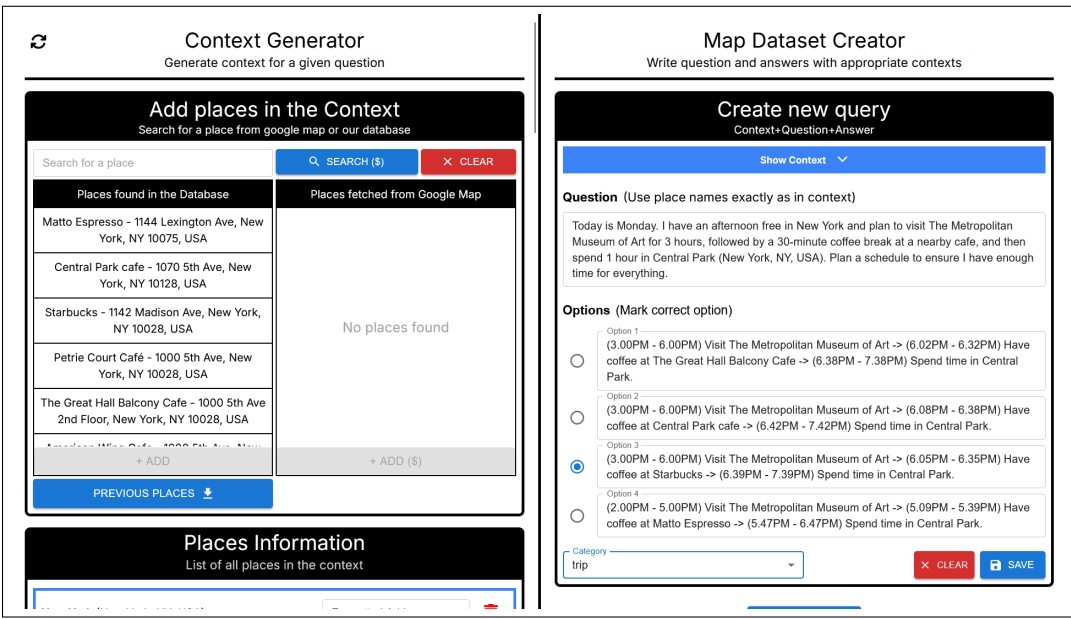

Figure 6: Screenshot of our Annotator Interface: MapQaTor

## C EVALUATION DETAILS

### C.1 PSEUDO-GOOGLE MAPS ENVIRONMENT

To ensure consistency between annotation and evaluation, a pseudo-Google Maps environment was developed with the following features:

- Caching: Information for over 13,000 locations was cached using Google Maps place_ids during both annotation and evaluation stages, ensuring consistency across updates. Table 8 presents the number of data entries for each API tool in our database

- API Simulation: A proxy interface mimics actual API interactions, enabling controlled testing while maintaining dynamic map-like attributes (e.g., travel times and place lists).

- Key-Query Mapping: Discrepancies between user queries and database keys were handled by storing all data using standardized place_ids obtained via a real API call.

This method maintains a static evaluation environment to preserve answer validity while simulating real-world API interactions by controlling dynamic variables like travel times, place attributes, and nearby location lists, which often change in live settings.

### C.2 MAPEVAL-TEXTUAL EVALUATION

In this evaluation setting, we provide the LLM with a pre-fetched context containing detailed information about specific locations, such as opening hours, distances between points of interest, and nearby amenities. The context is designed to simulate a real-world scenario.

| Tool | Entries (#) |
|---|---|
| PlaceDetailsTool | 13,354 |
| TravelTimeTool | 1,142 |
| DirectionsTool | 317 |
| NearbySearchTool | 481 |

Table 8: Number of data entries in the database for each API tool

Listing 4 demonstrates an example of this evaluation process. The context includes details about The Metropolitan Museum of Art, including its location, opening hours, and nearby cafes. The query asks for a time-optimized schedule that includes a 3-hour visit to the museum, followed by a 30-minute coffee break at a nearby cafe, and 1 hour spent in Central Park.

The available options offer different schedules, and the models are tasked with selecting the most appropriate one based on the provided context. As illustrated, models like Claude-3.5-Sonnet, Gemini-1.5-Pro, and GPT-4o correctly identify Option 3 as the best fit, considering the opening hours of each location and the feasible travel times between them. In contrast, Gemma-2.0-9B selects an incorrect option, indicating a misunderstanding of the cafe's closing hours.

This pre-fetched context evaluation allows us to test the model's ability to reason over structured information and make contextually informed decisions. It highlights the importance of understanding spatial relationships, operating hours, and timing constraints, all of which are crucial in real-world trip planning tasks.

## C.3 MAPEVAL-API EVALUATION

In this evaluation approach, we leverage a Zero Shot React Agent, which utilizes a dynamic tool-based framework to enhance the model's ability to respond to user queries effectively. Listing 6 illustrates the structured system prompt guiding the agent in employing various available tools. This framework allows the agent to access a range of functionalities, including retrieving place IDs, obtaining detailed information about locations, and estimating travel times between points of interest.

The Zero Shot React Agent's dynamic capabilities enable it to interact with tools in a systematic manner, ensuring accurate and contextually relevant responses. For example, the agent can utilize the PlaceId tool to obtain the unique identifier for a specified location, which can then be employed in subsequent actions, such as fetching detailed information with PlaceDetails or finding nearby places using NearbyPlaces. This modular approach not only simplifies complex queries but also grounds responses in real-time data.

The prompt structure encourages the agent to think critically about each step, starting with the user's question and leading to a carefully considered action. It determines the appropriate tool to use, specifies the necessary input, and provides a well-structured JSON blob for the action. The observation of the tool's output informs the next steps, allowing for iterative refinement of the response.

By employing this Zero Shot React Agent framework, we can assess the model's proficiency in utilizing external tools to generate accurate, contextually aware responses, ultimately enhancing its effectiveness in real-world applications.

## C.4 MAPEVAL-VISUAL EVALUATION

In this scenario, we provide Large Language Models (LLMs) with a map snapshot that offers critical geospatial context necessary for answering the query. This snapshot includes a default map view with clearly labeled locations and roads, aiding in the understanding of spatial relationships.

The evaluation process is demonstrated in Listing 13. The input consists of two parts: an image as visual context and a corresponding query with multiple answer options. In the example provided, the visual context includes a map displaying several golf clubs and a complex roadway network. The options represent possible answers that the evaluated models can choose from. The listing also shows the responses of various models, along with their explanations.

| Tool Name | Parameters | Description |
|---|---|---|
| PlaceSearch | placeName, placeAddress | Given a place name with address our tool calls Text Search API to get a list of places. Then choose the top place among them and returns its place id. |
| PlaceDetails | placeId | Given a place id our tool first searches in our database if not found, then uses Place Details API to fetch the details of the place. |
| TravelTime | originId, destinationId, travelMode | Given the place id of origin and destination, and travel mode our tool first searches in our database the duration (+distance) to go from origin to destination by preferred travel mode. If not found then queries Distance Matrix API. |
| Directions | originId, destinationId, travelMode | Given the place id of origin and destination, and travel mode our tool first searches in our database the available routes to go from origin to destination by preferred travel mode. If not found then queries Directions API. |
| NearbySearch | location, type, rankby, radius | This tool requires the place id of the place around which to retrieve place information. Additionally, the type of places, the order in which results are listed and distance within which to return place results. It then searches the database for stored Nearby Places. If absent, it queries Nearby Search API. |

Table 9: Summary of the tools used in evaluation.

Models like Gemini-1.5-Pro, GPT-4o-mini, and Claude-3.5-Sonnet successfully answered the query by correctly interpreting the geospatial information. On the other hand, Qwen-2VL-Chat selected an incorrect option, highlighting its difficulty in understanding spatial distances, which led to an erroneous answer.

This evaluation underscores the importance of providing visual context when testing LLMs' geospatial reasoning capabilities. By leveraging such visual aids, we can better assess how well these models understand and process spatial relationships in real-world scenarios

## D   FOUNDATION MODELS' DETAILS

Tables 10 and 11 provide comprehensive details of the open-source models utilized for dataset evaluation.

## E   FINE-GRAINED QUALITATIVE ERROR ANALYSIS

### MAPEVAL-TEXTUAL

*Commonsense Reasoning*: (i) Consider a scenario where the context states, "{Place_A} serves dinner, lunch, vegetarian food." When asked, "Does {Place_A} serve breakfast?" many LLMs respond, "There is not enough information in the context to answer," instead of simply saying "No." A human would deduce that since breakfast is not listed, {Place_A} does not serve it. (ii) Another challenge arises in planning questions. Even when opening hours are included in the context, LLMs may plan schedules that overlook constraints, such as visiting during closed hours while satisfying other

| Model | Parameters | Context Window |
|---|---|---|
| Phi-3.5-mini-instruct | 3.8B | 128K |
| Mistral-Nemo-Instruct-2407 | 7B | 128k |
| Mixtral-8x7B-Instruct-v0.1 | 7B | 32K |
| Qwen2.5-7B-Instruct | 7B | 128K |
| Qwen2.5-14B-Instruct | 14B | 128K |
| Qwen2.5-72B-Instruct | 72B | 128K |
| Llama-3.1-8B-Instruct | 8B | 128k |
| Llama-3.1-70B-Instruct | 70B | 128k |
| Llama-3.2-3B-Instruct | 3B | 128k |
| Llama-3.2-90B-text-preview | 90B | 128k |
| gemma-2-27b-it | 27B | 8.2k |
| gemma-2-9b-it | 9B | 8.2k |

Table 10: LLM model scales.

| Model | Parameters | Context Window |
|---|---|---|
| MiniCPM-Llama3-V-2_5 | 7B | 8.2k |
| Qwen2-VL-7B-Instruct | 8B | 32K |
| Llama-3-VILA1.5-8B | 8B | 8.2k |
| glm-4v-9b | 4.9B | 100k |
| InternLm-xcomposer2 | 7B | 96K |
| paligemma-3b-mix-224 | 3B | - |
| DocOwl1.5 | 8B | - |
| llava-v1.6-mistral-7b-hf | 7B | - |
| llava-1.5-7b-hf | 7B | - |

Table 11: VLM model scales.

conditions. For instance, although {Place_A} is open from 9:00AM to 3:00PM, the model might schedule a visit at 5:00PM, possibly due to inadequate training for this scenario.

*Spatial Reasoning:* (i) LLMs particularly struggle with queries requiring the calculation of spatial relationships, such as cardinal directions, straight-line distances, nearest points of interest (POIs), or step-by-step route planning. For example, in Place Info, Nearby, and Routing questions, examining 50 random questions that required such computations we observed a 10% decreased accuracy than others. This decline highlights the limitations of even dominant models like Gemini, which struggle with straight-line distance and direction calculations from geo-spatial data. (ii) LLMs also encounter difficulty with our domain specific questions that involve maths even in counting, especially when the count is large. For instance, in a query like "How many nearby restaurants have at least a 4.5 rating?", LLMs often fail to provide an accurate count.

*Temporal Reasoning:* LLMs struggle with temporal reasoning, which affects their performance on tasks like trip planning that require time manipulation. For example, when asked, "I want to visit A, B, and C. What is the most efficient order to visit?" the model must calculate travel times and determine the optimal route but often fails. Similarly, in a query like, "I want to visit A for 1 hour. What is the latest time I can leave home?" the model needs to subtract the visit duration and travel time from A's closing time, yet frequently makes errors in these simple time calculations.

## MAPEVAL-API

*Incorrect Tool Usage by Agents:* LLM-based agents often exhibit varying degrees of errors when utilizing map tools/APIs, particularly impacting Nearby queries. This task requires a complex set of arguments, and misinterpretation or improper use of these parameters frequently leads to failures in retrieving accurate results.

*Agents Stuck in Infinite Loops:* Invalid actions and repetitive loops contribute significantly to errors, especially in Routing queries. When there are no valid routes between an origin and destination, agents often fail to reconsider their approach or stop the process. Instead, they repeatedly attempt the same query with the same parameters, resulting in a deadlock and preventing progress.

**MAPEVAL-VISUAL**

*Spatial Reasoning:* In the Nearby category, models often exhibit confusion when multiple POIs are visually close together, leading to incorrect location selections. This indicates a struggle with fine-grained spatial analysis, affecting their ability to provide reliable responses and emphasizing the need for improved spatial awareness mechanisms.

*Temporal Reasoning:* In Routing queries, determining the fastest route requires detailed analysis of the source and destination, as well as transportation paths. VLMs often struggle with these calculations, resulting in a noticeable decline in performance and underscoring the difficulties in processing geographical information effectively.

*Detecting and Counting:* Models often struggle to accurately identify and count POIs in map images. For instance, when asked, "How many shopping stores or malls are there?" many proprietary VLMs may count incorrectly, with Claude-Sonnet listing an ATM as a store, leading to overcounting. Conversely, they sometimes undercount, (e.g., detecting only 6 to 8 out of an actual 12 malls).

# F QUALITATIVE EXAMPLES

| Type | Task | Question Example |
|---|---|---|
| Place Info | Textual/API | Which coffee shop is situated between Louvre Museum and Eiffel Tower? |
| | | What is the direction and straight-line distance from Victoria Falls to Hwange National Park? |
| | Visual | I'm at Baridhara K Block, feeling unwell, and need some medicine. What is a nearby pharmacy with a good rating that is open? |
| Nearby | Textual/API | How many shopping malls are there within a 500 m radius of Berlin Cathedral? |
| | | I am at Toronto Zoo. Today is Sunday and it's currently 8:30 PM. How many nearby ATMs are open now? |
| | Visual | I'm currently staying at Hörselberg-Hainich, while my friend is staying at Tüngeda. After we meet up, I want to visit an amusement park nearby. Can you suggest one that's close to us? |
| Routing | Textual/API | I want to walk from D03 Flame Tree Ridge to Aster Cedars Hospital, Jebel Ali. Which walking route involves taking the pedestrian overpass? |
| | | On the driving route from Hassan II Mosque to Koutoubia via A3, how many roundabouts I will encounter in total? |
| | Visual | Which restaurant is on the left side of the route from Metro El Golf to Metro Tobalaca L1? |
| Unanswerable | Textual/API | How many food stalls are there north of the overbridge at Hakaniemi? |
| | | Find a good coffee shop on the left side of my driving path from my home near Petaling Jaya to my office in Kuala Lumpur. |
| | Visual | How much time it would take to go to Igreja Nossa Senhora Da Conceição do Coroadinho - Matriz? |
| Trip | Textual/API | I live in Indira Road. At tomorrow 2 pm I will leave my house. I need to go to Military Museum to visit with friends for 2 hours and Multiplan Center to buy a keyboard (which will take 20 minutes) and Sonali Bank, BUET to receive my check book (which will take 30 minutes). In which order I should visit the places so that I reach there on time and come back home as early as possible. I will use public transport. |
| Counting | Visual | How many restaurants or clubs are on the bottom side of Linnakatu road? |

Table 12: Additional Complex Examples

Listing 1: Example evaluation of MAPEVAL-TEXTUAL `Green` : Correct Answer. `Red` : Wrong Answer.

```
Context:
Information of Eiffel Tower:
- Location: Av. Gustave Eiffel, 75007 Paris, France(48.8584, 2.2945).

Information of Mont Saint-Michel:
- Location: 50170 Mont Saint-Michel, France(48.6361, -1.5115).

Query:
What is the straight-line distance between the Eiffel Tower in Paris, France, and the Mont
Saint-Michel in Normandy, France?

Prompt:
Please respond in the following JSON format:
{
  "option_no": <option index>, // "option_no" refers to the number corresponding to the chosen
    answer from the list of options. It should be between 1 and 4.
  "explanation": "<reason>"
}

Example Prompt:
Question: What is the capital of France?
Option1: Berlin
Option2: Paris
Option3: Madrid
Option4: Rome

Example Response:
{
  "option_no": 2,
  "explanation": "Paris is the capital of France."
}

Provide your answer in this format. Remember to answer the question strictly based on the
given context, without using any external knowledge or assumptions.

Options:
1. 230 km
2. 250 km
3.  280 km
4. 310 km

Calude-3.5-Sonnet:
To answer this question, I'll need to calculate the straight-line distance between the two
points using their given coordinates. I'll use the Haversine formula for this calculation.
Eiffel Tower: (48.8584, 2.2945) Mont Saint-Michel: (48.6361, -1.5115) After performing the
calculation, the approximate distance is 284 km. Based on this result, the closest option is
280 km.
Gemini-1.5-Pro:
I cannot calculate the straight-line distance between the Eiffel Tower and Mont Saint-Michel
without using external knowledge or tools. This calculation requires geographic data and
mathematical formulas.
GPT-4o:
The straight-line distance between the coordinates of the Eiffel Tower (48.8584, 2.2945) and
Mont Saint-Michel (48.6361, -1.5115) is approximately 280 km.
Gemma-2.0-9B:
To determine the straight-line distance, we'd need to use the coordinates provided and apply a
 distance formula. Unfortunately, I can't perform calculations. **Therefore, I cannot answer
the question based on the given context.**
```

Listing 2: Example evaluation of MAPEVAL-TEXTUAL `Green` : Correct Answer. `Red` : Wrong Answer.

```
Context:
Information of Seurasaari Open-Air Museum:
- Location: Seurasaari, 00250 Helsinki, Finland(60.1856, 24.8846).

Information of Helsinki Central Station:
- Location: Kaivokatu 1, 00100 Helsinki, Finland(60.1719, 24.9414).
```

```
Query:
What is the direction of the Seurasaari Open-Air Museum from Helsinki Central Station?
```

```
Prompt:
Please respond in the following JSON format:
{
  "option_no": <option index>, // "option_no" refers to the number corresponding to the chosen
    answer from the list of options. It should be between 1 and 4.
  "explanation": "<reason>"
}

Example Prompt:
Question: What is the capital of France?
Option1: Berlin
Option2: Paris
Option3: Madrid
Option4: Rome

Example Response:
{
  "option_no": 2,
  "explanation": "Paris is the capital of France."
}

Provide your answer in this format. Remember to answer the question strictly based on the
given context, without using any external knowledge or assumptions.
```

```
Options:
1. East
2.  West
3. North
4. South
```

```
Calude-3.5-Sonnet:
To determine the direction of Seurasaari Open-Air Museum from Helsinki Central Station, I'll
compare their coordinates: Seurasaari Open-Air Museum: 60.1856, 24.8846 Helsinki Central
Station: 60.1719, 24.9414 Analyzing the coordinates: 1. Latitude: Seurasaari (60.1856) is
slightly north of Helsinki Central Station (60.1719) 2. Longitude: Seurasaari (24.8846) is
west of Helsinki Central Station (24.9414) The primary direction is west, with a slight
northward component.
Gemini-1.5-Pro:
Seurasaari Open-Air Museum (60.1856, 24.8846) is located at a higher latitude and almost same
longitude compared to Helsinki Central Station (60.1719, 24.9414). Higher latitude corresponds
 to North.
GPT-4o:
Seurasaari Open-Air Museum is located to the north of Helsinki Central Station based on their
coordinates.
Gemma-2.0-9B:
Seurasaari Open-Air Museum is located at 60.1856, 24.8846 and Helsinki Central Station is
located at 60.1719, 24.9414. The museum's coordinates are further north.
```

Listing 3: Example evaluation of MAPEVAL-TEXTUAL `Green`: Correct Answer. `Red`: Wrong Answer.

```
Context:
Nearby Restaurants of St. Lawrence Market are:
1. A&W Canada
- Open: Monday:   Open 24 hours , Tuesday: Open 24 hours, Wednesday: Open 24 hours, Thursday:
Open 24 hours, Friday: Open 24 hours, Saturday: Open 24 hours, Sunday: Open 24 hours.
....
7. Quik Sushi
- Open: Monday:   9:00AM-7:00PM , Tuesday: 11:30AM-7:30PM, Wednesday: 9:00AM-7:00PM, Thursday:
 9:00AM-7:00PM, Friday: 9:00AM-7:00PM, Saturday: 9:00AM-7:00PM, Sunday: 9:00AM-7:00PM.
....
9. Big Pita
- Open: Monday:   10:30AM-9:00PM , Tuesday: 10:30AM-9:00PM, Wednesday: 10:30AM-9:00PM,
Thursday: 10:30AM-9:00PM, Friday: 10:30AM-9:00PM, Saturday: 10:30AM-9:00PM, Sunday: 10:30AM
-9:00PM.
....
14. Pizza Pizza
- Open: Monday:   11:00AM-1:00AM , Tuesday: 11:00AM-1:00AM, Wednesday: 11:00AM-1:00AM,
Thursday: 11:00AM-1:00AM, Friday: 11:00AM-2:00AM, Saturday: 11:00AM-2:00AM, Sunday: 11:00AM
-1:00AM.
....
17. Market Street Catch
- Open: Monday:   11:00AM-8:00PM , Tuesday: 11:00AM-9:00PM, Wednesday: 11:00AM-9:00PM,
Thursday: 11:00AM-9:00PM, Friday: 11:00AM-10:00PM, Saturday: 11:00AM-10:00PM, Sunday: 11:00AM
-8:00PM.
....
20. Churrasco Of The St Lawrence
- Open: Monday: Closed, Tuesday: 7:00AM-7:00PM, Wednesday: 7:00AM-7:00PM, Thursday: 7:00AM
-7:00PM, Friday: 7:00AM-7:00PM, Saturday: 7:00AM-7:00PM, Sunday: 10:00AM-5:00PM.

Query:
How many nearby restaurants of St. Lawrence Market are open on Monday?

Prompt:
Please respond in the following JSON format:
{
  "option_no": <option index>, // "option_no" refers to the number corresponding to the chosen
   answer from the list of options. It should be between 1 and 4.
  "explanation": "<reason>"
}
Example Prompt:
Question: What is the capital of France?
Option1: Berlin, Option2: Paris, Option3: Madrid, Option4: Rome
Example Response:
{
  "option_no": 2,
  "explanation": "Paris is the capital of France."
}
Provide your answer in this format. Remember to answer the question strictly based on the
given context, without using any external knowledge or assumptions.

Options:
1. 2
2. 3
3.4
4.  5

Calude-3.5-Sonnet:
Based on the given information, 3 nearby restaurants are open on Monday. These are: A&W Canada
  (open 24 hours), Quik Sushi (open 9:00 AM     7:00 PM), and Big Pita (open 10:30 AM      9:00
PM). All other listed restaurants are either closed on Monday or their Monday hours are not
specified in the provided information.
Gemini-1.5-Pro:
Based on the provided information, these restaurants are open on Monday: A&W Canada, Quik
Sushi, Pizza Pizza, and Market Street Catch.
GPT-4o:
Only A&W Canada and Quik Sushi are open on Monday.
Gemma-2.0-9B:
Only A&W Canada is open 24 hours on Monday.
```

1296

1297 Listing 4: Example evaluation of MAPEVAL-TEXTUAL Green : Correct Answer. Red : Wrong
1298 Answer.

1299

Context:

1300 Information of The Metropolitan Museum of Art:
1301 - Location: 1000 5th Ave, New York, NY 10028, USA.
1302 - Open: Monday: 10:00AM-5:00PM, Tuesday: 10:00AM-5:00PM, Wednesday: Closed, Thursday: 10:00AM
     -5:00PM, Friday: 10:00AM-9:00PM, Saturday: 10:00AM-9:00-PM, Sunday: 10:00AM-5:00PM.

1303

1304 ....

1305 Travel Time from The Metropolitan Museum of Art to American Wing Cafe on foot is 5 mins (0.3
     km).
1306
1307 Travel Time from The Metropolitan Museum of Art to The Great Hall Balcony Cafe on foot is 2
     mins (0.1 km).
1308

1309 ....

1310 Nearby Cafes of The Metropolitan Museum of Art are (sorted by distance in ascending order):
1311 1. American Wing Cafe (1000 5th Avenue, New York)
1312 2. The Great Hall Balcony Cafe (1000 5th Avenue 2nd Floor, New York)
1313 3. Petrie Court Cafe (1000 5th Avenue, New York)
     4. Starbucks (1142 Madison Avenue, New York)
1314 .....

Query:

1315 Today is Monday. I have an afternoon free in New York and plan to visit The Metropolitan
1316 Museum of Art for 3 hours, followed by a 30-minute coffee break at a nearby cafe, and then
1317 spend 1 hour in Central Park (New York, NY, USA). Plan a schedule to ensure I have enough time
      for everything. I will walk between places.
1318

1319 Prompt:

1320 Please respond in the following JSON format:
     {
1321   "option_no": <option index>, // "option_no" refers to the number corresponding to the chosen
1322    answer from the list of options. It should be between 1 and 4.
       "explanation": "<reason>"
1323 }
     Example Prompt:
1324 Question: What is the capital of France?
1325 Option1: Berlin, Option2: Paris, Option3: Madrid, Option4: Rome
     Example Response:
1326 {
1327   "option_no": 2,
       "explanation": "Paris is the capital of France."
1328 }
1329 Provide your answer in this format. Remember to answer the question strictly based on the
1330 given context, without using any external knowledge or assumptions.

1331 Options:

1332 1. (3.00PM - 6.00PM) Visit The Metropolitan Museum of Art → (6.02PM - 6.32PM) Have coffee at
1333 The Great Hall Balcony Cafe → (6.38PM - 7.38PM) Spend time in Central Park.
     2. (3.00PM - 6.00PM) Visit The Metropolitan Museum of Art → (6.08PM - 6.38PM) Have coffee at
1334  Central Park cafe → (6.42PM - 7.42PM) Spend time in Central Park.
1335 3. (3.00PM - 6.00PM) Visit The Metropolitan Museum of Art → (6.05PM - 6.35PM) Have coffee at
      Starbucks  → (6.39PM - 7.39PM) Spend time in Central Park.
1336 4. (2.00PM - 5.00PM) Visit The Metropolitan Museum of Art → (5.09PM - 5.39PM) Have coffee at
1337 Matto Espresso → (5.47PM - 6.47PM) Spend time in Central Park.

1338

Calude-3.5-Sonnet:

1339 Option 3 is the most suitable schedule based on the given information. The Metropolitan Museum
1340  of Art is open on Mondays from 10:00 AM to 5:00 PM, so visiting from 3:00 PM to 6:00 PM is
     possible. Starbucks is open until 7:30 PM on Mondays, making it a viable option for a coffee
1341 break after the museum visit. Central Park is open until 1:00 AM, so spending time there from
1342 6:39 PM to 7:39 PM is feasible.

1343 Gemini-1.5-Pro:
     Option3 is the only option that has time slots within the opening hours of all the places
1344 mentioned. The Metropolitan Museum of Art is open until 5:00PM on Monday, Starbucks is open
1345 until 7:30PM and Central Park is open until 1:00AM.

1346 GPT-4o:
     Option 3 is feasible as Starbucks is open on Monday and is nearby. The schedule allows for a
1347 3-hour visit to The Metropolitan Museum of Art, a 30-minute coffee break, and 1 hour in
1348 Central Park.

     Gemma-2.0-9B:
1349 Option 1 provides the most feasible schedule. It allows for a 3-hour visit to the museum, a
     30-minute coffee break at The Great Hall Balcony Cafe (which is within walking distance), and
     a 1-hour visit to Central Park, all while ensuring enough time for travel between locations.

Listing 5: Example evaluation of MAPEVAL-TEXTUAL Green : Correct Answer. Red : Wrong Answer.

Context:
```
There are 3 routes from Petaling Jaya to Federal Territory of Kuala Lumpur by car. They are:
1. Via Damansara Link/Lebuhraya SPRINT/Sistem Penyuraian Trafik Kuala Lumpur Barat/E23 | 21
mins | 15.2 km
- Head east on Persiaran Tropicana toward Persiaran Tropicana
....
3. Via Damansara Link/Lebuhraya SPRINT/Sistem Penyuraian Trafik Kuala Lumpur Barat/E23 and
Jalan Maarof | 25 mins | 15.2 km
- Head east on Persiaran Tropicana toward Persiaran Tropicana
....
Nearby Cafes of Petaling Jaya are (sorted by distance in ascending order):
1. Dareia Coffee
- Address: Ps2-13 Lumi Tropicana, Persiaran Tropicana, Tropicana, Petaling Jaya.
- Rating: 5. (7 ratings).
....
Nearby Cafes of Federal Territory of Kuala Lumpur are (sorted by distance in ascending order):
1. Gwaku Coffee
- Address: ULG Lobby Menara TH 1 Sentral, Jalan Stesen Sentral 5, Kuala Lumpur Sentral, Kuala
Lumpur.
- Rating: 4.2. (62 ratings).
....
```

Query:
```
Find a good coffee shop on the left side of my driving path from my home near Petaling Jaya to
 my office in Kuala Lumpur.
```

Prompt:
```
Please respond in the following JSON format:
{
  "option_no": <option index>, // "option_no" refers to the number corresponding to the chosen
    answer from the list of options. It should be between 0 and 4.
  "explanation": "<reason>"
}

Example Prompt:
Question: What is the capital of France?
Option0: Unanswerable, Option1: Berlin, Option2: Paris, Option3: Madrid, Option4: Rome
Example Response:
{
  "option_no": 2,
  "explanation": "Paris is the capital of France."
}
Provide your answer in this format. Remember to answer the question strictly based on the
given context, without using any external knowledge or assumptions.
```

Options:
0. Unanswerable
1. Dareia Coffee
2. Gwaku Coffee
3. The Lemon Tree
4. Coffeeloca

Calude-3.5-Sonnet:
```
The Lemon Tree is a good coffee shop option on the left side of the driving path from Petaling
 Jaya to Kuala Lumpur. It is located at the Ground Floor of Menara Lien Hoe, which is along
Persiaran Tropicana. This cafe is on the initial part of the journey and has a rating of 4 out
 of 5 based on 836 ratings, indicating it's a popular and well-liked spot. It's also described
 as having a 'Moderate' price level, making it a suitable choice for a coffee stop on the way
to the office.
```
Gemini-1.5-Pro:
```
There is no information on which side of the driving path the cafes are located.
```
GPT-4o:
```
Gwaku Coffee is located on Jalan Stesen Sentral 5, which is on the driving path from Petaling
Jaya to Kuala Lumpur.
```
Gemma-2.0-9B:
```
Gwaku Coffee is located in Kuala Lumpur Sentral, which is on the driving path from Petaling
Jaya to Kuala Lumpur.
```

## Listing 6: MAPEVAL-API System Prompt

```
Respond to the human as helpfully and accurately as possible. You have access to the following
  tools:

PlaceSearch: Get place ID for a given location., args: {'placeName': {'description': 'Name of
the place', 'title': 'Placename', 'type': 'string', 'placeAddress': {'description': 'Address
of the place', 'title': 'Placeaddress', 'type': 'string'}}}

PlaceDetails: Get details for a given place ID., args: {'placeId': {'description': 'Place Id
of the location', 'title': 'Placeid', 'type': 'string'}}

NearbySearch: Get nearby places around a location., args: {'placeId': {'description': 'The id
of the place around which to retrieve nearby places.', 'title': 'Placeid', 'type': 'string'},
'type': {'description': 'Type of place (e.g., restaurant, hospital, etc). Restricts the
results to places matching the specified type.', 'title': 'Type', 'type': 'string'}, 'rankby':
 {'default': 'distance', 'description': 'Specifies the order in which places are listed.
Possible values are: (1. prominence (default): This option sorts results based on their
importance. When prominence is specified, the radius parameter is required. 2. distance: This
option sorts places in ascending order by their distance from the specified location. When
distance is specified, radius is disallowed. In case you are not concerned about the radius,
use rankby as distance.)', 'title': 'Rankby', 'type': 'string'}, 'radius': {'anyOf': [{'type':
 'integer'}, {'type': 'null'}], 'default': None, 'description': 'Defines the distance (in
meters) within which to return place results.', 'title': 'Radius'}}

TravelTime: Estimate the travel time between two places., args: {'originId': {'description': '
Place Id of Origin', 'title': 'Originid', 'type': 'string'}, 'destinationId': {'description':
'Place Id of Destination', 'title': 'Destinationid', 'type': 'string'}, 'travelMode': {'
description': 'Mode of transportation (driving, walking, bicycling, transit)', 'title': '
Travelmode', 'type': 'string'}}

Directions: Get directions/routes between two places., args: {'originId': {'description': '
Place Id of Origin', 'title': 'Originid', 'type': 'string'}, 'destinationId': {'description':
'Place Id of Destination', 'title': 'Destinationid', 'type': 'string'}, 'travelMode': {'
description': 'Mode of transportation (driving, walking, bicycling, transit)', 'title': '
Travelmode', 'type': 'string'}}

Use a json blob to specify a tool by providing an action key (tool name) and an action_input
key (tool input).

Valid "action" values: "Final Answer" or PlaceSearch, PlaceDetails, NearbySearch, TravelTime,
Directions

Provide only ONE action per $JSON_BLOB, as shown:

```
{
  "action": $TOOL_NAME,
  "action_input": $INPUT
}
```

Follow this format:

Question: input question to answer
Thought: consider previous and subsequent steps
Action:
```
$JSON_BLOB
```
Observation: action result
... (repeat Thought/Action/Observation N times)
Thought: I know what to respond
Action:
```
{
  "action": "Final Answer",
  "action_input": "Final response to human"
}
```

Begin! Reminder to ALWAYS respond with a valid json blob of a single action. Use tools if
necessary. Respond directly if appropriate. Format is Action:```$JSON_BLOB```then Observation
:.
Thought:
```

Listing 7: Example evaluation of MAPEVAL-API `Green` : Correct Answer. `Red` : Wrong Answer.

```
Query:
Can I visit The Great Pyramid of Giza at 5:00 PM Saturday?
_____

Options:
1. Yes
2.  No
_____

Calude-3.5-Sonnet:

To answer this question accurately, I'll need to check the opening hours of The Great Pyramid
of Giza. Let me start by getting the place ID for the location.

Action:
```
{
  "action": "PlaceSearch",
  "action_input": {
    "placeName": "The Great Pyramid of Giza",
    "placeAddress": "Al Haram, Nazlet El-Semman, Al Giza Desert, Giza Governorate, Egypt"
  }
}
```

Observation: ChIJGymPrIdFWBQRJCSloj8vDIE
Thought:Now that I have the place ID for The Great Pyramid of Giza, I'll use it to get more
details about the location, including its opening hours.

Action:
```
{
  "action": "PlaceDetails",
  "action_input": {
    "placeId": "ChIJGymPrIdFWBQRJCSloj8vDIE"
  }
}
```

Observation:
- Location: Al Haram, Nazlet El-Semman, Al Haram, Giza Governorate 3512201, Egypt (29.9791705,
 31.1342046).
- Open: Monday: 8:00AM-4:30PM, Tuesday: 8:00AM-4:30PM, Wednesday: 8:00AM-4:30PM, Thursday:
8:00AM-4:30PM, Friday: 8:00AM-4:30PM, Saturday: 8:00AM-4:30PM, Sunday: 8:00AM-4:30PM.
- Rating: 4.6. (27048 ratings).

Thought:Based on the information provided by the PlaceDetails tool, I can now answer the
question.

The Great Pyramid of Giza is open on Saturdays from 8:00 AM to 4:30 PM. The question asks if
it can be visited at 5:00 PM on Saturday. Since 5:00 PM is after the closing time of 4:30 PM,
the answer is No.

Therefore, the correct option is Option 2: No.

Action:
```
{
  "action": "Final Answer",
  "action_input": "2"
}
```
```

1512

1513 Listing 8: Example evaluation of MAPEVAL-API Green : Correct Answer. Red : Wrong Answer.

1514

```
Query:
```
1515 I am currently staying at Khaosan Tokyo Origami in Asakusa, Tokyo. I want to visit the Senso-
1516 ji Temple for 1 hour, Shibuya Crossing for 1.5 hours, Ueno Park for 2 hours, and the Tokyo
1517 Skytree for 1.5 hours. I have 3.5 hours available. I will leave my hostel at 7 am. How many
    places can I visit? I will walk between places.
1518 ─────

1519 Options:
    1. One
1520 2. Two
1521 3. Three
1522 4. Four
    ─────
1523 Calude-3.5-Sonnet:
1524 To answer this question accurately, I'll need to calculate the travel times between these
    locations and the time spent at each place. Let's break this down step by step.
1525

1526 First, I'll need to get the place IDs for all the locations mentioned.

    Action:
1527 ```
1528 {
1529   "action": "PlaceSearch",
      "action_input": {
1530     "placeName": "Khaosan Tokyo Origami",
1531     "placeAddress": "Asakusa, Tokyo"
      }
1532 }
1533 ```

1534 .....

1535
    Observation: ChIJ35ov0dCOGGARKvdDH7NPHX0
1536 Thought:Thank you for providing the place ID for Tokyo Skytree. Now that we have all the place
     IDs, let's calculate the travel times between these locations using the walking mode of
1537 transportation.
1538

1539 Let's start with the travel time from Khaosan Tokyo Origami to Senso-ji Temple.

1540 Action:
1541 ```
    {
1542   "action": "TravelTime",
1543   "action_input": {
        "originId": "ChIJ4ehN78GOGGARyvT_Bjj5s-c",
1544     "destinationId": "ChIJ8T1GpMGOGGARDYGSgpooDWw",
1545     "travelMode": "walking"
      }
1546 }
1547 ```

1548
    Response: [[{'distance': {'text': '0.3 km', 'value': 267}, 'duration': {'text': '4 mins', '
1549 value': 227}, 'status': 'LOCAL'}]]

1550
    .....
1551
    Observation: Travel Time on foot is 51 mins (3.5 km).
1552 Thought:Now that we have all the travel times and the time to be spent at each location, let's
1553  calculate how many places can be visited within the 3.5 hours available, starting at 7 am.

1554 1. Khaosan Tokyo Origami to Senso-ji Temple: 4 minutes
1555 2. Time at Senso-ji Temple: 1 hour
    3. Total time so far: 1 hour 4 minutes
1556
1557 At this point, it's 8:04 am, and we've used 1 hour 4 minutes of the available 3.5 hours.

1558 The next destination, Shibuya Crossing, is 2 hours 52 minutes away on foot, which would exceed
1559  the remaining time (2 hours 26 minutes) even before spending any time there.

1560 Therefore, the traveler can only visit one place (Senso-ji Temple) within the given time
1561 constraints.

1562 Action:
1563 ```
1564 {
      "action": "Final Answer",
1565   "action_input": "Option1"
    }
    ```

Listing 9: Example evaluation of MAPEVAL-VISUAL `Green` : Correct Answer. `Red` : Wrong Answer.

`Context:`

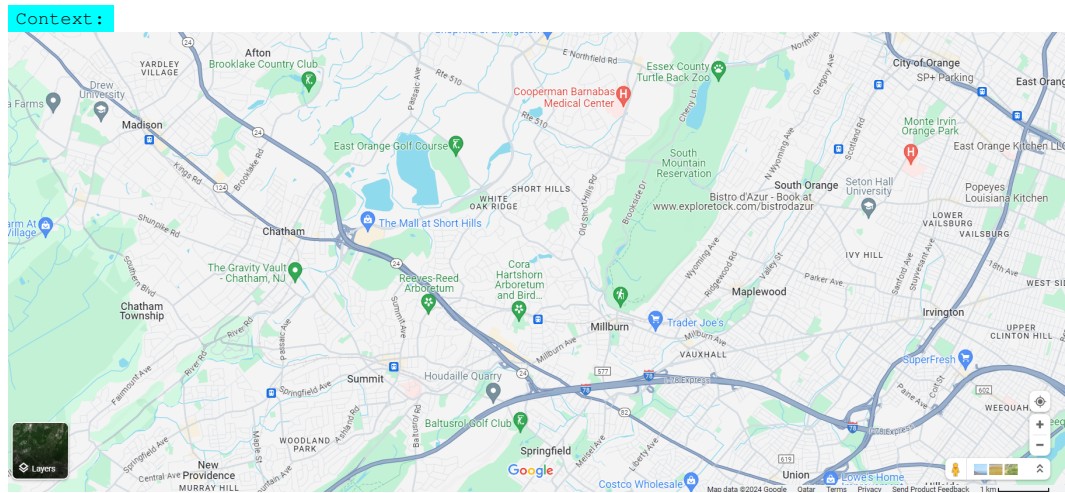

`Query:`

Which Golf Clubis located at the intersection of Springfield  and Houdaille Quarry?

`Prompt:`

Choose the correct option from the image also show the answer with visible quotation and give proper explanation.

`Options:`

1.Brooklake Country Club
`2.` Baltusrol Golf Club
3.East Orange Golf Course
4.Reeves-Reed Arboretum

`Gemini-1.5-Pro:`

The golf club located at the intersection of Springfield and Houdaille Quarry is "Baltusrol Golf Club".
You can clearly see the label for Baltusrol Golf Club positioned right above Springfield Ave, with the golf flag icon situated directly at the point where Springfield Ave intersects with Houdaille Quarry.

`GPT-4o-mini:`

The correct option is "2. Baltusrol Golf Club."
Explanation:
In the image, the location of Houdaille Quarry and Springfield can be found near the center at the junction of major roads. The Baltusrol Golf Club is marked nearby, indicating it is situated at that intersection. The other options, such as Brooklake Country Club, East Orange Golf Course, and Reeves-Reed Arboretum, do not appear to be located at the intersection of Springfield and Houdaille Quarry.

`Calude-3.5-Sonnet:`

Based on the map image, the correct answer is: "2. Baltusrol Golf Club"
Explanation:
The map shows Baltusrol Golf Club located near the intersection of Springfield Avenue and Houdaille Quarry. You can see "Baltusrol Golf Club" marked on the map just south of where Route 24 meets I-78, which is very close to where Springfield Avenue and Houdaille Quarry ( both visible on the map) would intersect. Therefore, Baltusrol Golf Club is the golf club located at the intersection of Springfield Avenue and Houdaille Quarry.

`Llama-3-VILA1.5-8B:`

The answer is 3.East Orange Golf Course. The location of East Orange Golf Course is at the intersection of Springfield and Houdaille Quarry, as shown in the image. The image is a screenshot of a map, and the location of East Orange Golf Course is marked with a red pin.

Listing 10: Example evaluation of MAPEVAL-VISUAL `Green` : Correct Answer. `Red` : Wrong Answer.

`Context:`

`Query:`

I am presently visiting  Mount Royal Park . Could you please inform me about the nearby historical landmark?

`Prompt:`

Choose the correct option from the image also show the answer with visible quotation and give proper explanation.

`Options:`

`1.` Circle Stone
2.secret pool
3.Maison William Caldwell Cottingham
4.Poste de cavalerie du Service de police de la Ville de Montreal

`Gemini-1.5-Pro:`
The correct answer is "Maison William Caldwell Cottingham".
The map shows the location of "Mount Royal Park". Just to the east of that marker is the "Maison William Caldwell Cottingham". The other options are located further away from Mount Royal Park.

`GPT-4o-mini:`
The correct option is:"3. Maison William Caldwell Cottingham"
Explanation:
Maison William Caldwell Cottingham is a historical landmark located near Mount Royal Park. It reflects the architectural heritage and history of the area, providing insights into the past. The other options, while interesting, do not hold the same historical significance as Maison William Caldwell Cottingham.

`Calude-3.5-Sonnet:`
Based on the map image provided, the correct answer is:"Circle Stone"
Explanation:
The question asks about a nearby historical landmark to Mount Royal Park. Looking at the map, we can see Mount Royal Park clearly labeled. Among the given options, "Circle Stone" is the closest historical landmark visible on the map near Mount Royal Park. It's marked with a pin icon just north of the park area.
The other options are either farther away or not visible on this particular map view:
- "secret pool" is visible but seems to be farther to the northeast.
- "Maison William Caldwell Cottingham" is visible but located to the southeast of the park.
- "Poste de cavalerie du Service de police de la Ville de Montreal" is visible but located to the west of the park.

`Llama-3-VILA1.5-8B:`
The answer is 3.Maison William Caldwell Cottingham.
Maison William Caldwell Cottingham is a historic house museum located in Montreal, Quebec, Canada. It is located in the Mount Royal Park, which is a popular tourist attraction in Montreal. The house was built in 1851 and is a fine example of Gothic Revival architecture. It is now a museum that showcases the history and culture of Montreal..

Listing 11: Example evaluation of MAPEVAL-VISUAL `Green` : Correct Answer. `Red` : Wrong Answer.

`Context:`

`Query:`

Which school is closed to Club Atletico Puerto Comercial?

`Prompt:`

Choose the correct option from the image also show the answer with visible quotation and give proper explanation.

`Options:`

`0.` Unanswerable
1. Secondary school No. 25
2. PROVINCIAL SCHOOL No. 67 TOMAS ESPORA
3. Province of Buenos Aires SCHOOL No. 507
4. SCHOOL No.5 Province of Buenos Aires

`Gemini-1.5-Pro:`

The answer is "0) Unanswerable".
The map provided does not show any schools. Therefore, it is impossible to determine which school is closed to Club Atletico Puerto Comercial based solely on the provided image.

`GPT-4o-mini:`

To determine which school is closest to Club Atletico Puerto Comercial, we need to analyze the locations of the listed schools in relation to the club's position on the map.
Analysis:
- Club Atletico Puerto Comercial is located near the center bottom of the map.
- The other schools must be found or deduced based on their general proximity to this location
From the visible landmarks and overall arrangement on the map, PROVINCIAL SCHOOL No. 67 TOMAS ESPORA appears to be the closest to Club Atletico Puerto Comercial.

`Calude-3.5-Sonnet:`

Based on the information provided in the image, the correct answer is:0) Unanswerable
Explanation:
The map does not show any schools or educational institutions near Club Atletico Puerto. While it shows various landmarks and businesses, including "Club Atletico Puerto", there is no information about any schools visible on this map.
The options provided mention specific schools, but none of these are shown or labeled on the map image.
Given the limited information in the image and the absence of any school locations, the question cannot be answered based solely on what's shown here, making "Unanswerable" the most appropriate choice.

`Llama-3-VILA1.5-8B:`

The answer is 1) Secondary school No. 25.
The image shows a map of Puerto Comercial, Argentina, with various locations marked. One of the locations marked is a secondary school, which is the closest school to Club Atl tico Puerto Comercial. The answer is 1) Secondary school No. 25.

Listing 12: Example evaluation of MAPEVAL-VISUAL Green : Correct Answer. Red : Wrong Answer.

Context:

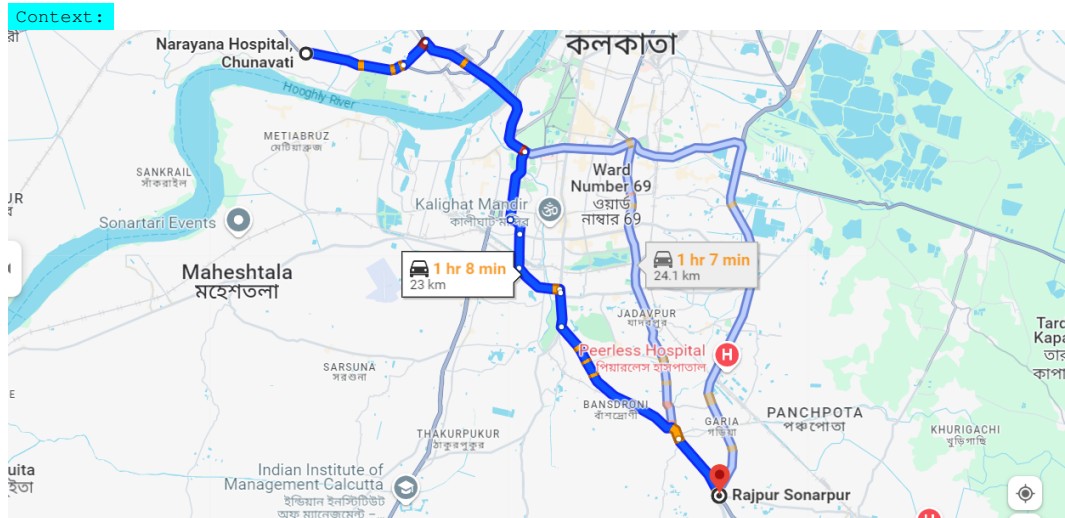

Query:
How far is Rajpur Sonarpur in West Bengal, India, from Narayana Hospital, Chunavati, Andul Rd, Howrah, West Bengal, if I take the following route:
Continue on Andul Rd (42 m, 13 sec).
Follow Andul Rd and NH 12 to Alipore Rd in Alipore, Kolkata (32 min, 11.3 km).
Continue on Alipore Rd to Barhans (32 min, 9.5 km).
Continue straight past Union Bank ATM-Garia onto Garia Main Rd/Netaji Subhash Chandra Bose Rd (7 min, 2.1 km).
Pass by Jharna Ghee And Honey on the right.

Prompt:
Choose the correct option from the image also show the answer with visible quotation and give proper explanation.

Options:
1. 23 km
2. 23.2km
3. 24.1km
4. 27.6km

Gemini-1.5-Pro:
The image clearly shows the total distance as "23 km".
Therefore, the correct answer is "1) 23 km".

GPT-4o-mini:
Based on the provided information and the image, the distance from Narayana Hospital, Chunavati, Howrah, to Rajpur Sonarpur is approximately 23 km.
The answer is option 1) "23 km"
Explanation:
- The route consists of several parts, with the total distance calculated and displayed in the map:
- The total time shown on the map indicates a travel duration of around 1 hour and 8 minutes for the distance of 23 km.

Calude-3.5-Sonnet:
Based on the information provided in the image, the correct answer is:"23 km"
Explanation:
The map shows a route from Narayana Hospital, Chunavati (which is near the Hooghly River in the northwest of the map) to Rajpur Sonarpur (in the southeast of the map). The route is highlighted in blue. In the bottom left corner of this blue route, there's a distance and time estimate clearly visible:"1 hr 8 min 23 km"

Llama-3-VILA1.5-8B:
The answer is 27.6km.
The distance from Narayana Hospital, Chunavati, Andul Rd, Howrah, West Bengal, to Rajpur Sonarpur is 27.6 kilometers. The route you described involves traveling along Andul Rd, NH 12, Alipore Rd, Garia Main Rd/Netaji Subhash Chandra Bose Rd, and passing by Jharna Ghee And Honey on the right. The total distance covered is 27.6 kilometers.

Listing 13: Example evaluation of MAPEVAL-VISUAL Green : Correct Answer. Red : Wrong Answer.

Context:

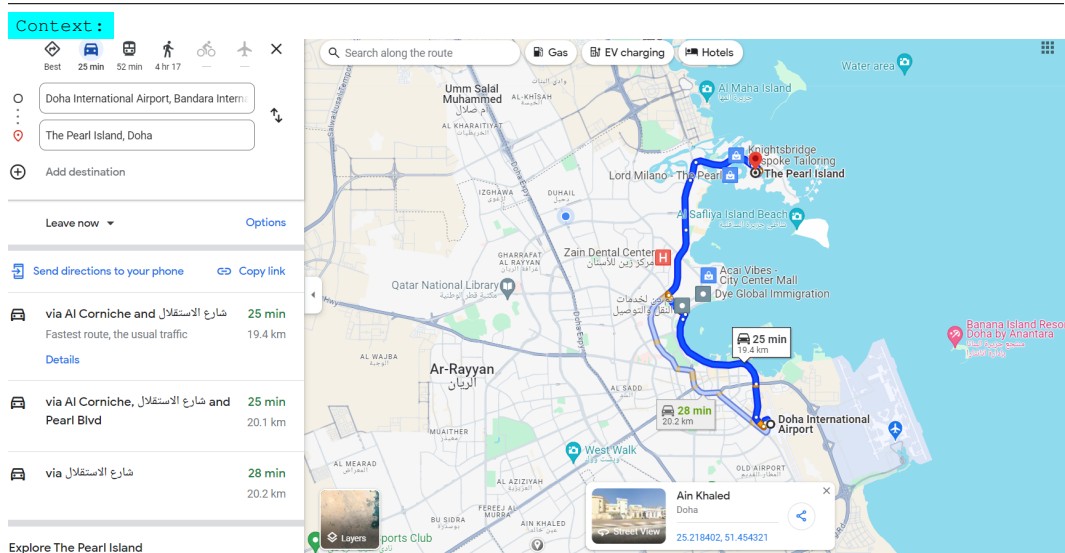

Query:
```
If traveling at a constant speed of 5.4 km/h, how long will it take to travel from Point Doha
International Airport, Bandara Internasional Hamad, Qatar Air Traffic Control, MB l.p, Doha to
 Point The Pearl Island, Doha if the distance is 19.4 km?
```

Prompt:
```
Choose the correct option from the image also show the answer with visible quotation and give
proper explanation.
```

Options:
```
1. 3.12 Minutes
2. 3.72 Minutes
3. 3.82 Minutes
```
4. ` 3.99 Minutes`

Gemini-1.5-Pro:
```
The correct answer is "25 min".
The provided options in the question (3.12, 3.72, 3.82, 3.99 minutes) are nonsensical given
the travel time will take at least 25 minutes based on the provided map.
Explanation:
The image clearly shows that the estimated travel time from Doha International Airport to The
Pearl Island is **25 minutes**.  The provided speed of 5.4 km/h and distance of 19.4 km are
irrelevant in this case, as we have the estimated travel time provided directly from the map
software.
```

GPT-4o-mini:
```
1. Convert 3.59 to minutes:
3.59 hours = 3.59 * 60 = 215.4 minutes
As the distance of 19.4 km at 5.4 km/h actually corresponds to more than 3 minutes, none of
the provided answer choices seem directly correct.
If we need to choose the closest available option that might reflect an error in reasoning or
calculation leading to a proportional time in minutes.
```

Calude-3.5-Sonnet:
```
Distance = 19.4 km
Speed = 5.4 km/h
Time = 19.4 km / 5.4 km/h = 3.59 hours
     = 3.59 hours * 60 minutes/hour = 215.4 minutes
Looking at the given options, none of them are correct as they all show times in minutes that
are far too short.
```

Llama-3-VILA1.5-8B:
```
The answer is (1) 3.12 Minutesv.
The time it takes to travel 19.4 km at a constant speed of 5.4 km/h is 19.4 km / 5.4 km/h =
3.12 minutes.
```

## G DATASET STATISTICS AND ANALYSIS

| Country | Count | Country | Count | Country | Count |
|---|---|---|---|---|---|
| Bangladesh | 132 | United States | 57 | United Arab Emirates | 40 |
| India | 33 | Canada | 31 | United Kingdom | 27 |
| Japan | 24 | Australia | 19 | Pakistan | 16 |
| Qatar | 15 | Saudi Arabia | 12 | China | 12 |
| Germany | 10 | Argentina | 10 | Luxembourg | 9 |
| Italy | 8 | Spain | 8 | Brazil | 8 |
| South Africa | 7 | Poland | 7 | New Zealand | 7 |
| France | 7 | Denmark | 7 | Bhutan | 7 |
| Hungary | 7 | Czechia | 7 | Chile | 7 |
| Sierra Leone | 7 | Malaysia | 7 | Sweden | 7 |
| Norway | 7 | Peru | 6 | Colombia | 6 |
| Zimbabwe | 6 | Ireland | 6 | Mexico | 6 |
| Egypt | 6 | Greece | 6 | Austria | 6 |
| Indonesia | 6 | Nepal | 6 | Netherlands | 6 |
| Vietnam | 6 | Belgium | 6 | South Korea | 6 |
| Portugal | 6 | Morocco | 6 | Finland | 6 |
| Thailand | 6 | South Sudan | 6 | Russia | 6 |
| Switzerland | 6 | Turkey | 6 | Singapore | 6 |

Table 13: Distribution of Questions Across Countries.

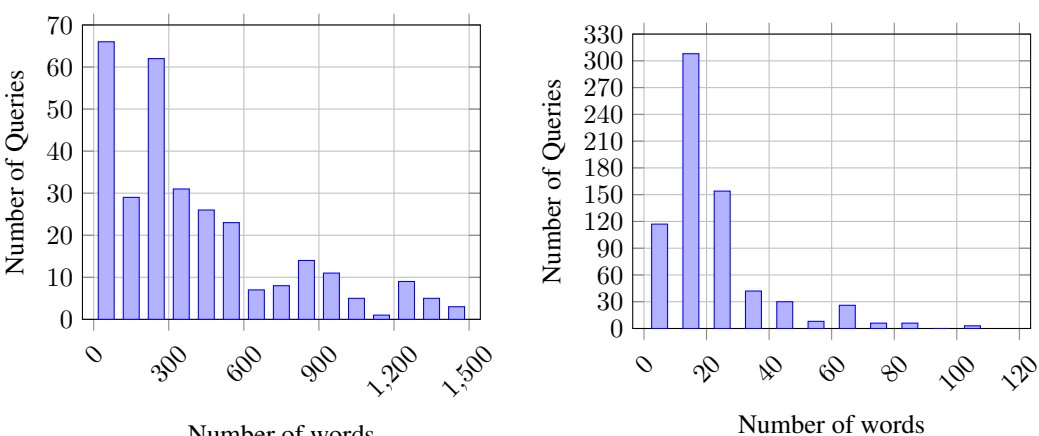

Figure 7: Distribution of Textual-Context Lengths (300 samples)

Figure 8: Distribution of Question Lengths (700 samples)

### G.1 ZOOM DETAILS

In our dataset, zoom levels range from 8.0 to 21.0, as shown in 11.Each visual context is paired with a Google Maps URL, such as https://www.google.com/maps/@35.7048455,139.763263,16.71z?entry=ttu, where the value before the "z" (e.g., 16.71) represents the zoom level. This allows us to easily extract zoom information directly from the URL, ensuring that each visual context can be accurately mapped to its respective level of detail.

## H EVALUATION RESULTS VISUALIZATION

In this section, we present the results of our evaluations through a series of charts that summarize the performance of different models across various categories. These visualizations provide a clear and

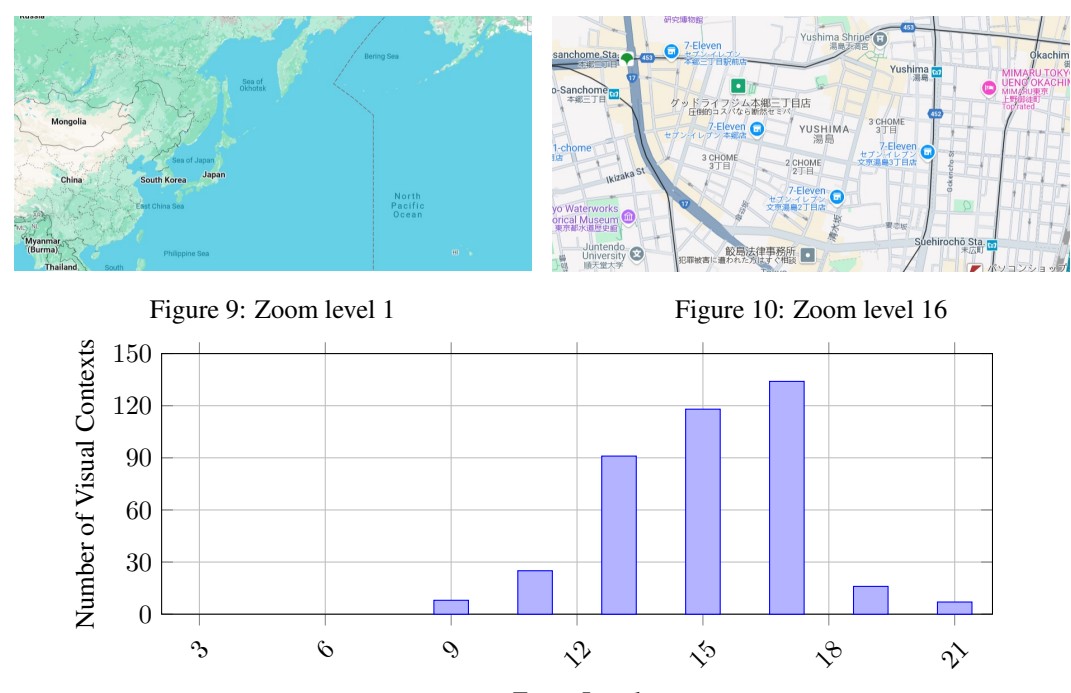

Figure 9: Zoom level 1        Figure 10: Zoom level 16

Figure 11: Distribution of Zoom Levels (400 Visual Samples)

concise comparison of model effectiveness in addressing textual, API-based, and visual geospatial queries. The charts are designed to highlight key trends, strengths, and limitations of the evaluated approaches.

## H.1 MAPEVAL-TEXTUAL

Figure 12 illustrates the performance of models on MAPEVAL-TEXTUAL.

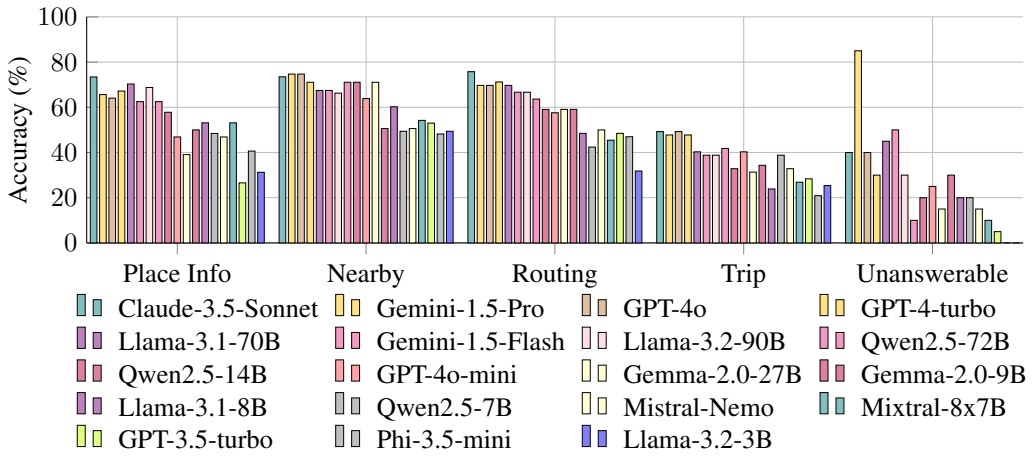

Figure 12: MAPEVAL-TEXTUAL categorical accuracy

Figure 13 illustrates how accuracy of models in MAPEVAL-TEXTUAL changes with different context length.

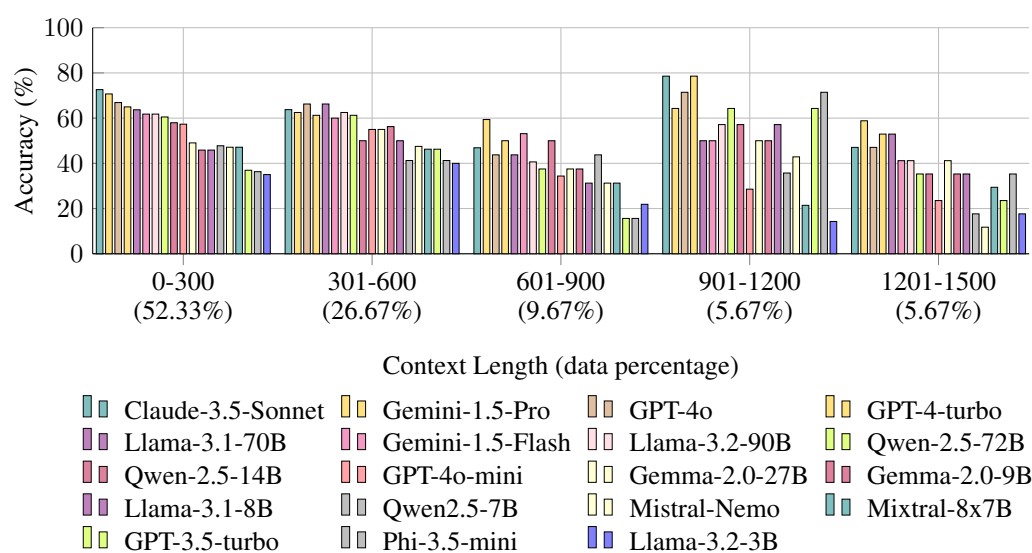

Figure 13: Accuracy vs. Context Length (MAPEVAL-TEXTUAL )

## H.2    MAPEVAL-API

Figure 14 illustrates the performance of models on MAPEVAL-API.

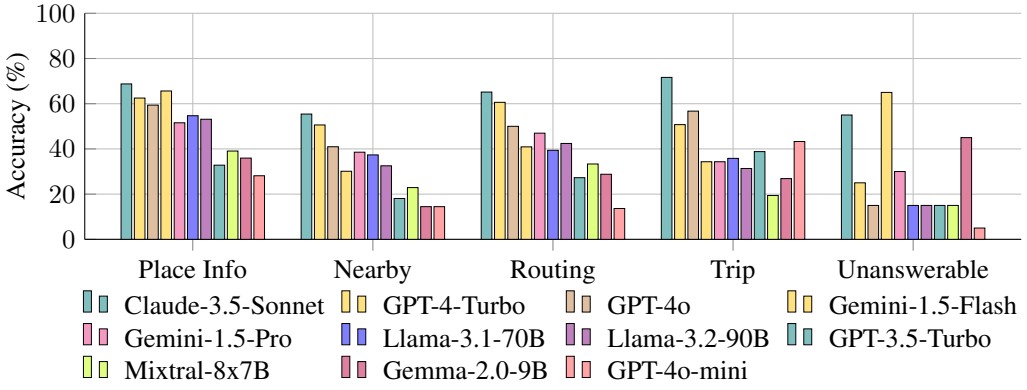

Figure 14: MAPEVAL-API categorical accuracy

Additionally in Figure 15, we can visualize the number of times agent stopped due to iteration limit. This happens when agents repeatedly calls the same api with same parameters and gets the same response. While GPT-3.5-Turbo encounters 16 infinite iterations, Claude-3.5-Sonnet doesn't face this issue.

## H.3    MAPEVAL-VISUAL

Figure 16 illustrates the performance of models on MAPEVAL-VISUAL.

# I    ADDITIONAL EXPERIMENT RESULTS

For additional experiments, we filtered questions from our textual/API dataset into three subcategories:

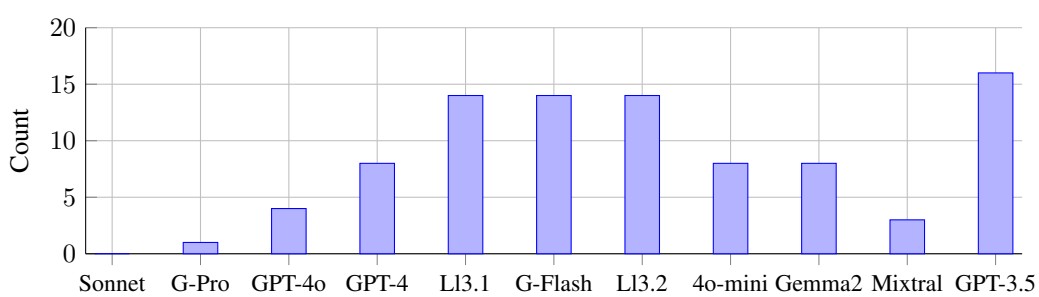

Figure 15: Number of times agent stopped due to iteration limit

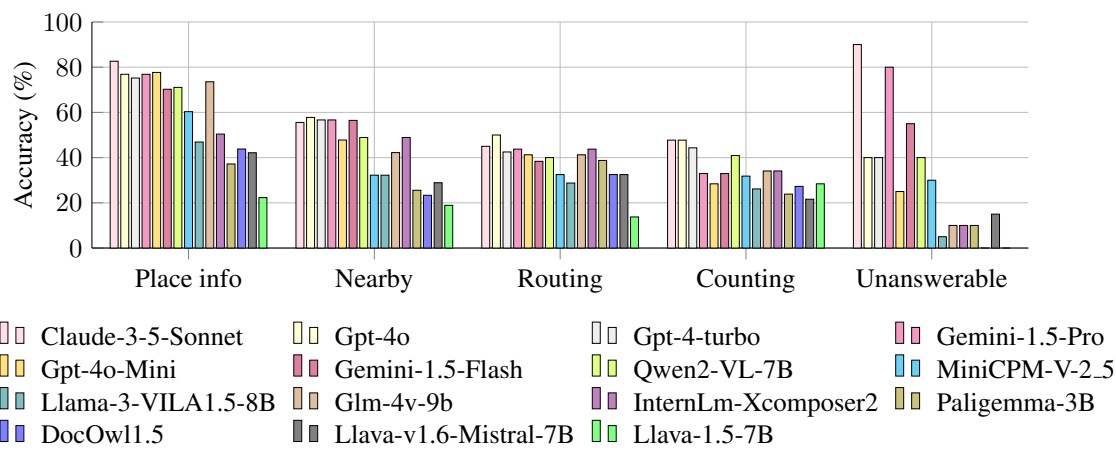

Figure 16: MAPEVAL-Visual categorical accuracy

1. *Straight-Line Distance* (47 questions): These questions require the computation of straight-line distances, such as "What is the straight-line distance between the Atomium in Brussels and the Belfry of Bruges?"

2. *Cardinal Direction* (24 questions): These involve determining cardinal directions[2], e.g., "What is the direction of the Little Mermaid statue from Copenhagen Central Station?"

3. *Counting* (23 questions): Questions involving counting entities, such as "How many convenience stores are there within a 400 m radius of the Tokyo Tower?"

We visualized the accuracy of various LLMs on these subcategories under the MAPEVAL-TEXTUAL setting, with the following findings:

1. *Straight-Line Distance:* Figure 17, illustrates the accuracy on straight-line distance related questions. We can see that all models struggled, with the best accuracy being only 51.06%.

2. *Cardinal Direction:* Figure 18, illustrates the accuracy on cardinal-direction related questions. Here LLMs showed significant variability. While Claude-3.5-Sonnet achieved 91% accuracy, Gemma-2.0-27B scored only 16.67%.

3. *Counting:* Figure 19, illustrates the accuracy on counting related questions. In this case, Claude-3.5-Sonnet underperformed compared to the open-source Gemma-2.0-27B (60.87% accuracy).

We identified a scope for improvement in these areas and enhanced the models' capabilities by integrating external tools (e.g., a calculator) specifically designed for calculating straight-line distances and cardinal directions. For straight-line distances, we employed the Haversine formula[3] to com-

---

[2]https://en.wikipedia.org/wiki/Cardinal_direction
[3]https://en.wikipedia.org/wiki/Haversine_formula

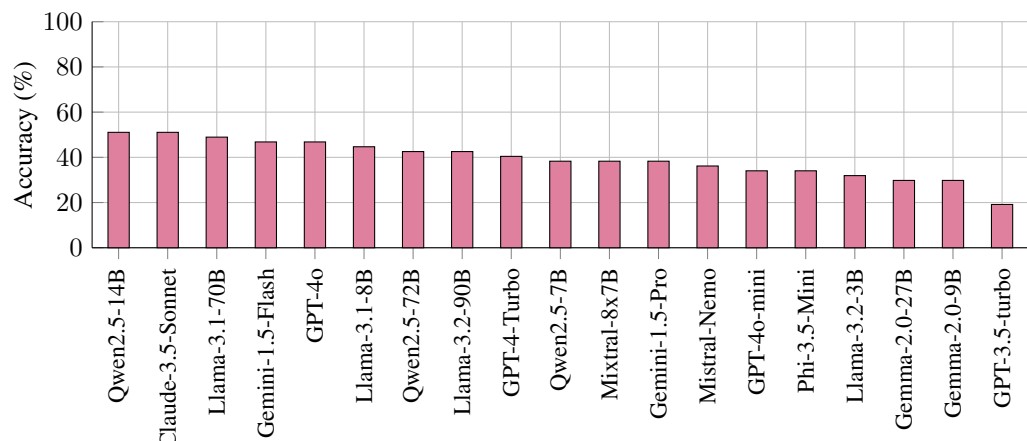

Figure 17: Accuracy of LLMs on questions which needs **Straight-Line Distance** computation

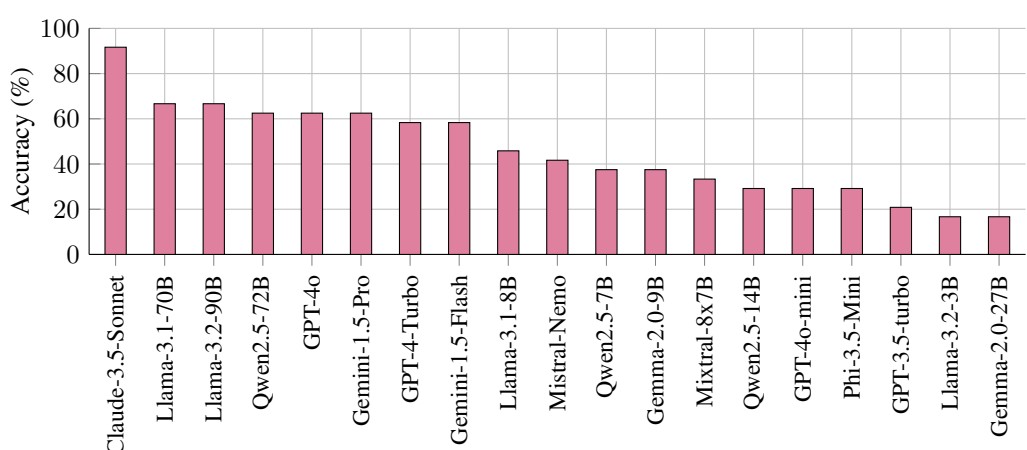

Figure 18: Accuracy of LLMs on questions which needs **Cardinal Direction** computation

pute the great-circle distance. To determine cardinal directions, we calculated the bearing[4] between two geographic coordinates. Figures 20 and 21 demonstrate a significant improvement in model performance with these tools.

For straight-line distance-related questions, the best accuracy jumped from 51.06% to 85.11%. Similarly, for cardinal-direction questions, the top model achieved an accuracy of 95.83%, compared to the previous maximum of 91.67%. In the case of GPT-4o-mini, these enhancements led to even further progress, with the model demonstrating a leap in both straight-line distance and cardinal direction accuracy, surpassing previous models. In the case of GPT-4o-mini, these enhancements led to even further progress, with the model demonstrating a remarkable leap in both straight-line distance and cardinal direction accuracy. Specifically, the straight-line distance accuracy improved from 34.04% to 78.72%, while cardinal-direction accuracy increased from 29.17% to 91.67%.

These results highlight the limitations of current LLMs in handling fine-grained geospatial queries independently and emphasize the value of augmenting LLM capabilities with external computational tools. Future work can explore the integration of more robust external services to address the nuances of spatial reasoning comprehensively.

---

[4]https://en.wikipedia.org/wiki/Bearing_(navigation)

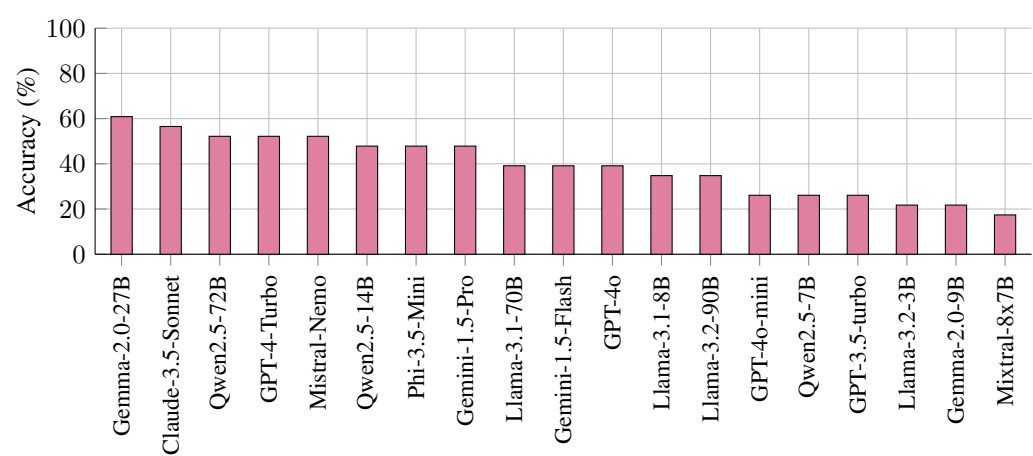

Figure 19: Accuracy of LLMs on **Counting** related questions

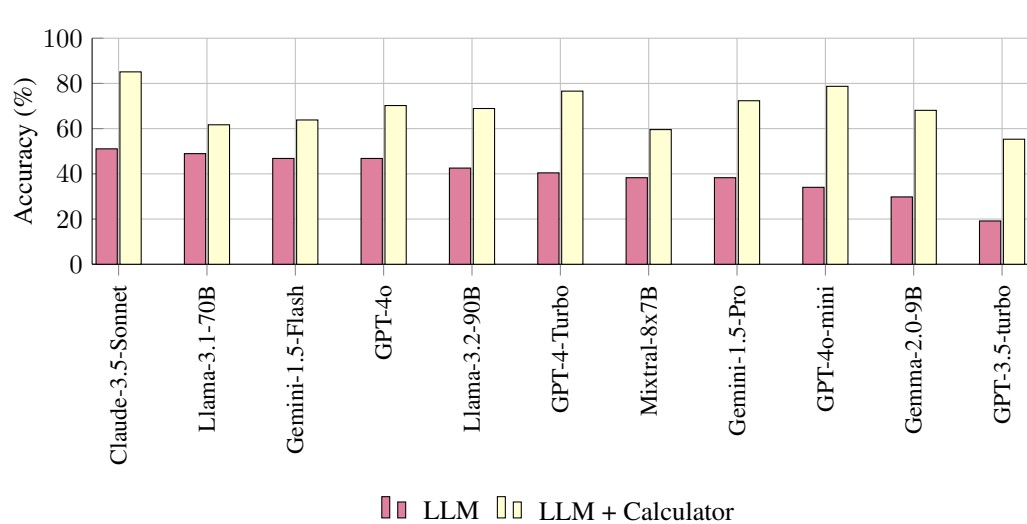

Figure 20: Improved accuracy of LLMs after integrating calculator to compute **Straight-Line Distance**

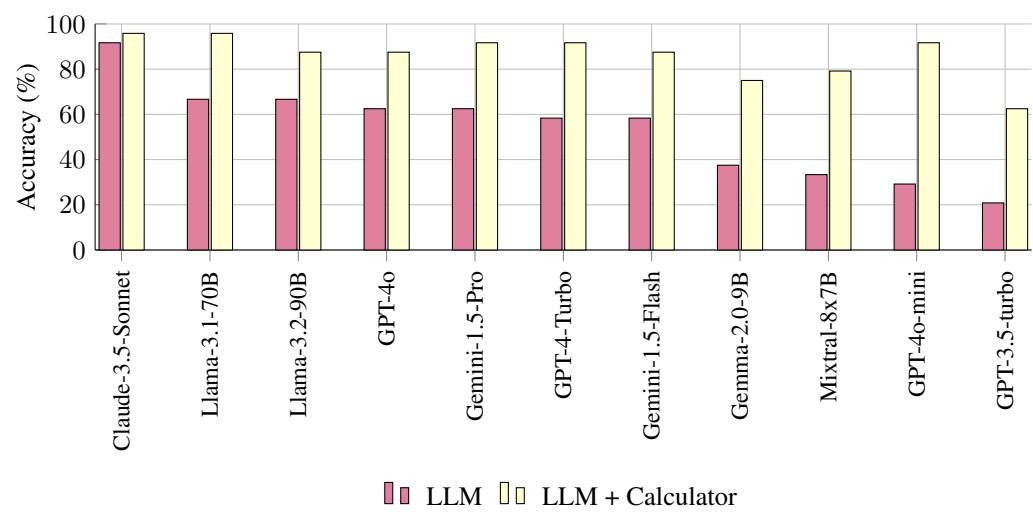

Figure 21: Improved accuracy of LLMs after integrating calculator to compute **Cardinal Direction**

