# OpenReview forum: "MapEval: A Map-Based Evaluation of Geo-Spatial Reasoning in Foundation Models"
_ICLR.cc/2025/Conference — Submitted to ICLR 2025_

### Official Review · Reviewer_WpVB · 2024-10-27

**Soundness:** 2
**Presentation:** 3
**Contribution:** 2
**Rating:** 5
**Confidence:** 3

**Summary:**

The authors claimed that the capabilities of the foundation model in location or map-based reasoning have not been systematically studied. This paper creates a benchmark MAPEVAL to assess map-based user queries with geo-spatial reasoning. MAPEVAL evaluates 25 well-known foundation models on textual, API-based, and visual three types of tasks and 550+ multiple-choice questions. Although, no single foundation model outperforms others on all tasks, they still analyze the strengths and weaknesses of current prominent models. Overall, the benchmarking of foundation model in map tasks like map search is useful, while more in-depth analysis would be beneficial. In addition, I am unsure about whether this paper is actually a fit for ICLR.

**Strengths:**

1. This paper introduces the MAPEVAL benchmark to evaluate the geo-spatial reasoning capabilities of foundation models in map-based user queries, which is a meaningful effort in general.
2. The paper is generally detailed and easy to follow.
3. The distinction between open-source and closed-source foundation models is useful as a reference for the future.

**Weaknesses:**

1. The number of test cases available for evaluating foundation models is relatively small, with only around 550 cases.
2. The benchmark framework is unable to evaluate multi-modal large language models that combine both text and visual information.
3. The analysis is not in-depth enough. It would be beneficial to point out the root causes of the performance (rank), which could inspire the future development of map-related foundation models and the gradual specialization of general-purpose models.

**Questions:**

1. The last paragraph of the introduction mentions ablation experiments, but I could not find the experimental content or results in the main text.
2. This paper compares the evaluation results of several foundation models, but it should further clarify whether any specific geographic specialized foundation models were used to highlight the specialization in geo-spatial reasoning tasks. If none were used, please explain the reasons. Example models are those developed with geographic knowledge like K2, and remote sensing models (the number is increasing).

---

> ### Author Response · Authors · 2024-11-23
> **Addressing W1**
>
> Thank you for your recognition of the applicability, effectiveness of our proposed benchmarking task, and comprehensive experiments in our work!
>
> > W1: The number of test cases available for evaluating foundation models is relatively small, with only around 550 cases.
>
> **(Also discussed in Response to Reviewer bA3F W2)**
>
> Due to the high cost of both foundation models as well as tools/APIs, recent language agents often tend to evaluate on a small number of sub-sampled dataset. For example ReACT [1]  uses only 500 random samples from AlfWorld dataset, similarly Reflexion often tends to use [2] only 100 examples from HotpotQA). Therefore, recently proposed tool oriented or intense reasoning benchmark datasets are found to reasonable in size in order to be cost-effective: API-Bank: [3] (400 instances)  Logical-reasoning benchmark LogiQA: [4] (641 examples), the most popular problem solving (code generation benchmarks) HumanEval: [5]  (164 instances only), CodeContests [6] (156 problems), Tao-bench [7] (165 problems), OS World [8] (369 problems), App world [9] (750 problems), TravelPlanner [10] (1.2K problems). Consequently, we carefully construct our problem instances balanced in size and covering different challenges.
>
> However, we thank you for the feedback and leveraging the rebuttal time-periods of ICLR, we strive to increase our problem set to 700 and ensure that each of the 54 countries is now represented by a minimum of 6 questions while maintaining the same quality control and pipeline. This improvement addresses the concern regarding insufficient representation of certain countries, making the evaluation more robust and geographically diverse. In our observation, further attempts in incorporating additional questions from different countries revealed no significant change in model performance. To maintain a cost-effective testbed, we opted not to include more instances, suggesting that our evaluation problem set already offers sufficient diversity for robust and reliable assessment. This improvement addresses the concern regarding insufficient representation of certain countries, making the evaluation more robust and geographically diverse.
>
> In addition, as outlined in Section 3.1, our goal is to benchmark on high standard real-user queries aiming to solve day-to-day actual Map application tasks and hence we do not scale up by synthesizing.
>
> **References:**
>
> [1] Yao, Shunyu, et al. "ReAct: Synergizing Reasoning and Acting in Language Models." The Eleventh International Conference on Learning Representations.\
> [2] Shinn, Noah, et al. "Reflexion: Language agents with verbal reinforcement learning." Advances in Neural Information Processing Systems 36 (2024).\
> [3] Li, Minghao, et al. "API-Bank: A Comprehensive Benchmark for Tool-Augmented LLMs." Proceedings of the 2023 Conference on Empirical Methods in Natural Language Processing. 2023.\
> [4] Liu, Jian, et al. "LogiQA: a challenge dataset for machine reading comprehension with logical reasoning." Proceedings of the Twenty-Ninth International Conference on International Joint Conferences on Artificial Intelligence. 2021.\
> [5] Chen, Mark, et al. "Evaluating large language models trained on code." arXiv preprint arXiv:2107.03374 (2021).\
> [6] Li, Yujia, et al. "Competition-level code generation with alphacode." Science 378.6624 (2022): 1092-1097.\
> [7] Yao, Shunyu, et al. "$\tau$-bench: A Benchmark for Tool-Agent-User Interaction in Real-World Domains." arXiv preprint arXiv:2406.12045 (2024).\
> [8] Xie, Tianbao, et al. "Osworld: Benchmarking multimodal agents for open-ended tasks in real computer environments." arXiv preprint arXiv:2404.07972 (2024).\
> [9] Trivedi, Harsh, et al. "AppWorld: A Controllable World of Apps and People for Benchmarking Interactive Coding Agents." Proceedings of the 62nd Annual Meeting of the Association for Computational Linguistics (Volume 1: Long Papers). 2024.\
> [10] Xie, Jian, et al. "TravelPlanner: A Benchmark for Real-World Planning with Language Agents". Poster presented at the Forty-First International Conference on Machine Learning (ICML), 2024.

---

> ### Author Response · Authors · 2024-11-23
> **Addressing W2, W3 and Q1**
>
> > W2: The benchmark framework is unable to evaluate multi-modal large language models that combine both text and visual information.
>
> We do not understand this point. As discussed in Fig 1, Section 3.2 and so on, our inputs in MapEval-vision task are both a textual query and a map snapshot image which you also have mentioned in your summary. Note that each MapEval-vision query includes a textual context (within ~100 tokens). In Appendix F, we depict full examples of several instances also (e.g, listing 12 on page 33). We have a complete result Section 4.2.3 evaluating  multi-modal VLMs in Table 5.  In addition, a subcategory of MapEval-vision queries called 'Extra Information Processing' involves map snapshot images containing OCR texts such as user ratings or detailed routes in trip history (mentioned in updated paper line 186) that must be processed to generate answers (see listing 13 in Appendix F).  How then it can not evaluate multi-modal models taking both text and image input.
>
> > W3: The analysis is not in-depth enough. It would be beneficial to point out the root causes of the performance (rank), which could inspire the future development of map-related foundation models and the gradual specialization of general-purpose models.
>
> Thank you for your valuable feedback. As a dataset and benchmark work, we have already provided comprehensive evaluation and analysis, detailing fine-grained model performances and identifying root causes of errors across various foundation models (discussed in Section 4.3). However, to address your concern for deeper analysis, we have expanded our study in response to your comments. We have now included a performance ranking across different reasoning sub-tasks (e.g., measuring straight-line distances, calculating cardinal directions, and solving counting problems) to better understand model strengths and weaknesses. Additionally, we have added a new section (Section 5 in the updated paper) with extended analyses. We also present complementary enhancement strategies that address identified model limitations, particularly in reasoning aspects where models lack explicit training. These findings can inform the development of more capable next-generation foundation models.
>
> > Q1: The last paragraph of the introduction mentions ablation experiments, but I could not find the experimental content or results in the main text.
>
> Thanks for pointing out this. This is a typo, we meant to say “our in-depth analysis” (updated in the revised paper).

---

> ### Author Response · Authors · 2024-11-23
> **Addressing Q2**
>
> > Q2: This paper compares the evaluation results of several foundation models, but it should further clarify whether any specific geographic specialized foundation models were used to highlight the specialization in geo-spatial reasoning tasks. If none were used, please explain the reasons. Example models are those developed with geographic knowledge like K2, and remote sensing models (the number is increasing).
>
> **Clarification on the Exclusion of K2 from Our Evaluations**
>
> Thank you for your comment. We would like to clarify the distinction between the models we used and those mentioned, particularly K2 and remote sensing models. K2, as outlined in the full title "K2: A Foundation Language Model for Geoscience Knowledge Understanding and Utilization," [1] is designed specifically for geoscience-related tasks, not geographic or geospatial reasoning tasks. It focuses on questions related to geoscience knowledge, such as “What kind of rock belongs to dynamic metamorphic rocks?”, which is distinct from the map-based user queries we address in this work.
>
> Given this difference in domain focus, we anticipated that K2 would not perform well for map-based questions. In response to your feedback, we evaluated K2, and the evaluation confirmed that its performance was very poor, with an accuracy of 20.33%. Given that K2 performed no better than random guessing, we decided not to include it in our evaluations. Besides, at the time of our evaluation, there were no geospatial specialized foundation models that met the specific requirements of our task.
>
> ---
>
> | Overall | Place Info | Nearby | Routing | Trip | Unanswerable |
> |:----------:|:----------:|:----------:|:----------:|:----------:|:----------:|
> |   20.33   | 25.00   | 20.48    | 15.15  | 20.90 | 20.00   |
>
> *Table 1: Performance of K2 in MapEval-Textual*
>
> ---
>
> **Explanation on Remote Sensing Models in Our Study**
>
> Regarding remote sensing models, it is important to note that our study does not involve remote sensing images. Instead, we provide digital map view snapshots (See Listing 9 - 13 in Appendix F for example snapshots) to Vision-Language Models (VLMs), which focus on interpreting map data rather than satellite or remote sensing images. As such, the task we are addressing is fundamentally different from remote sensing tasks, which is why we did not include remote sensing models in our evaluations. A more detailed comparison is added in Appendix A.3.
>
> We hope this explanation clarifies the rationale behind our model selection. Thank you again for your valuable feedback.
>
> **References:**
>
> [1] Deng, Cheng, et al. "K2: A foundation language model for geoscience knowledge understanding and utilization." Proceedings of the 17th ACM International Conference on Web Search and Data Mining. 2024.

---

> ### Author Response · Authors · 2024-11-28
> **Reminder**
>
> Dear reviewer,
>
> It is a gentle reminder to acknowledge our rebuttal and make necessary reassessment of the score. Again, we thank you for your time and effort in reviewing our work.

---

### Official Review · Reviewer_rmq7 · 2024-10-29

**Soundness:** 2
**Presentation:** 2
**Contribution:** 1
**Rating:** 3
**Confidence:** 4

**Summary:**

This paper presents the MAPEval evaluation dataset, which assesses the capabilities of foundational models (such as large language models and vision-language models) in spatial reasoning and geographic information extraction applications from three perspectives: textual, API-based, and visual evaluation modes. The authors provide a detailed description of the dataset creation process, introduce the performance of multiple models on the dataset, and thoroughly analyze the abilities of different models.

**Strengths:**

1. This paper is clear, with a detailed description of the dataset collection process and contribution points.
2. This paper tests the performance of multiple models on the dataset, which can provide guidance for readers on using different models.

**Weaknesses:**

(1) This paper introduces a dataset to evaluate the capabilities of existing models, but I didn't gain much insight from it. Specifically, I learned about the capabilities of different models on this dataset, but then what? It feels like an incomplete study. Perhaps this paper would be more suitable for a data track.
(2) Since the authors only collected a dataset and evaluated different models, the research lacks sufficient innovation. It appears more like an experimental report than a research paper.
(3) What I was hoping to see was some form of inspiration or insight from this paper, but unfortunately, I couldn’t find that. All I can gather is that different models may have advantages and disadvantages in different areas.
(4) I think the paper might be lacking in terms of analyzing the distinct capabilities demonstrated by different models based on the evaluation results. The authors should draw conclusions on how to design a model that performs well across all aspects. This would greatly interest readers and significantly enhance the research value of the paper.
(5) If the authors could attempt and verify the suggestions from point four, I think this paper would be very strong.
(6) In reality, I feel there's also a lack of discussion on the practical applications of large language models (LLMs) and visual language models (VLMs), as they tend to serve different purposes. Specifically, can the conclusions from this paper guide users on designing a comprehensive framework to solve real-world problems? I think further discussion in these areas would be beneficial.
In summary, while the contribution of this paper is clear, it indeed lacks certain elements that would make it more impactful.

**Questions:**

I believe this research could be further improved by discussing why different models perform differently on the collected dataset, and exploring the relationship between the existing results and model designs to provide more insights.

---

> ### Author Response · Authors · 2024-11-23
> **Addressing W1, W2 and W3**
>
> Thank you for your feedback. We understand that there may have been some confusion, and it seems that you might have interpreted our paper as an interpretability or methodology paper. As a result, some of your comments, such as "this paper should be a paper in the data track," were a bit surprising. We would like to clarify that this paper is indeed a submission under the primary area of "datasets and benchmarks" for ICLR 2025. Nonetheless, we appreciate your insights and address your comments below.
>
> > W1: This paper introduces a dataset to evaluate the capabilities of existing models, but I didn't gain much insight from it. Specifically, I learned about the capabilities of different models on this dataset, but then what? It feels like an incomplete study. Perhaps this paper would be more suitable for a data track.
>
> > W2: Since the authors only collected a dataset and evaluated different models, the research lacks sufficient innovation. It appears more like an experimental report than a research paper.
>
> >W3: What I was hoping to see was some form of inspiration or insight from this paper, but unfortunately, I couldn’t find that. All I can gather is that different models may have advantages and disadvantages in different areas.
>
> Given that this paper is submitted under the "Dataset and Benchmark" track, its primary focus is on introducing a benchmark dataset to evaluate the capabilities of existing large language models (LLMs) and vision-language models (VLMs). The key contribution lies in providing a standardized resource for assessing model performance in geospatial reasoning and location-based tasks, an area currently lacking robust benchmarks.
>
> The insights gained from evaluating models on this dataset are intended to (1) highlight existing gaps in model capabilities (discussed in Section 4.2),  (2) identify areas for improvement in geospatial reasoning (discussed in Section 4.2, 4.3, Appendix E) and (3) serve as a foundation for future research and model development (Section 4.2, 4.3, Appendix E and 5).
>
> While the evaluation itself demonstrates the utility of the dataset, the broader implications include facilitating advancements in multimodal, tool-oriented and geospatial AI through the availability of a comprehensive benchmark. We believe this aligns well with the objectives of the "Dataset and Benchmark" category, where the focus is on providing resources that can drive progress in AI research.
>
> We hope this clarification addresses your concerns and underscores the completeness of our study within the context of this track.

---

> ### Author Response · Authors · 2024-11-23
> **Addressing W4 and W5**
>
> > W4: I think the paper might be lacking in terms of analyzing the distinct capabilities demonstrated by different models based on the evaluation results. The authors should draw conclusions on how to design a model that performs well across all aspects. This would greatly interest readers and significantly enhance the research value of the paper.
>
> > W5: If the authors could attempt and verify the suggestions from point four, I think this paper would be very strong.
>
> Thank you for your valuable feedback. In response to your suggestion for a deeper analysis of the distinct capabilities demonstrated by different models, we have expanded our study to highlight their comparative strengths and weaknesses across various tasks in the benchmark. Our analyses cover a wide spectrum of possible enhancements for various aspects.
>
> For instance:
>
> - Claude-3.5-Sonnet excels in identifying cardinal directions, achieving an accuracy of 91%, while Gemma-2.0-27B has the lowest accuracy at 16.67% on this task.
> - All LLMs struggled with measuring straight-line distances, with the highest accuracy being only 51.06%.
> - In counting-related questions, the open-source Gemma-2.0-27B performed the best, achieving 60.87% accuracy, which underscores areas where other high-performing models, like Claude-3.5-Sonnet, could improve.
>
> Therefore, ensembling different LLMs and their dynamic deployment based on user queries can be a potential solution. Furthermore, we have investigated specific failure cases to identify distinct behavioral patterns. For example, GPT-3.5-Turbo encountered a deadlock 16 times during MapEval-API evaluations, while Claude-3.5-Sonnet did not face such issues.
>
> To address these issues, we extended model capabilities by providing access to external tools (e.g., calculator) specifically for calculating straight-line distances and cardinal directions. This resulted in a dramatic improvement, with accuracies increasing by over 50% in certain cases (see Table 6).
>
> For instance:
>
> - The accuracy of Claude-3.5-Sonnet in calculating straight-line distances increased from 51.06% to 85.11%, demonstrating the utility of integrating external tools.
>
> - GPT-4o-mini, which initially struggled with cardinal direction tasks, saw its performance increase from 29.17% to 91.67%, showcasing a remarkable transformation with tool support.
>
> - Even open-source models like Gemma-2.0-9B benefited, achieving an accuracy boost in straight-line distance tasks from 29.79% to 68.90%.
>
> These improvements highlight the challenges LLMs face when reasoning spatially without external support. By leveraging tools, models can offload computationally intensive or context-specific reasoning tasks, enabling more precise and reliable results.
>
> We also discuss challenges in API calling agents such as infinite loops (in our error analysis in Section 4.3) or further strategies for building enhanced agents in Section 5 such as using Chameleon [1].
>
> We have added the new experimental results and in-depth analysis in the revised paper. Refer to section 4.3 and Section 5 for details.
>
> **Reference:**
>
> [1] Lu, Pan, et al. "Chameleon: Plug-and-play compositional reasoning with large language models." Advances in Neural Information Processing Systems 36 (2024).

---

> ### Author Response · Authors · 2024-11-23
> **Addressing W6 and Q1**
>
> > W6: In reality, I feel there's also a lack of discussion on the practical applications of large language models (LLMs) and visual language models (VLMs), as they tend to serve different purposes. Specifically, can the conclusions from this paper guide users on designing a comprehensive framework to solve real-world problems? I think further discussion in these areas would be beneficial.
>
> In the introduction (Section 1), we discuss the motivation behind our work, which is grounded in real users' daily interactions with map services. Foundation models serve as a bridge between map services (APIs) and recommendation systems, enhancing user experience. Potential real-life applications could include copilots map service, AI assistants in customer service, or location-based web applications. We believe our benchmarking dataset will play a pioneering role in advancing research in this area.
>
> > Q1: I believe this research could be further improved by discussing why different models perform differently on the collected dataset, and exploring the relationship between the existing results and model designs to provide more insights.
>
> Thanks for the valuable feedback. In this “dataset and benchmark” work, with comprehensive evaluation and analysis, we present fine-grained model performances and identify various root causes of their mistakes and study each foundation model performs in fine-grained categories. We also present complementary ways to enhance (such as in Section 4 particularly in 4.3 and Section 5 (newly added)) which can be inferred as identified limitations or weaknesses of models i.e., potentially on which reasoning aspects the models are not explicitly trained for. We believe these findings can help the development of more capable next generation foundation models. However, even for open-source foundation models, they rarely release full training data and procedure (i.e., in fact they are open-weight models) which limits us from further  causal analysis studies. Even so, we believe such further studies are beyond our scope of this paper which we leave as future work.

---

> ### Author Response · Authors · 2024-11-28
> **Reminder**
>
> Dear reviewer,
>
> It is a gentle reminder to acknowledge our rebuttal and make necessary reassessment of the score. Again, we thank you for your time and effort in reviewing our work.

---

### Official Review · Reviewer_bA3F · 2024-11-02

**Soundness:** 3
**Presentation:** 3
**Contribution:** 3
**Rating:** 5
**Confidence:** 4

**Summary:**

The paper introduces MAPEVAL, a benchmark designed to assess the geo-spatial reasoning capabilities of foundation models in map-based scenarios. MAPEVAL evaluates models across three task types—textual, API-based, and visual—that involve querying and reasoning with map-based information. These tasks test a model’s ability to handle complex geographical contexts, such as travel routes, location details, and map infographics.

MAPEVAL’s dataset includes 550+ multiple-choice questions spanning 138 cities and 54 countries, examining models’ abilities in areas like spatial relationships, navigation, and planning. The evaluation showed that top models, such as Claude-3.5-Sonnet, GPT-4o, and Gemini-1.5-Pro, performed well overall but still fell significantly short of human accuracy, particularly in complex tasks requiring deep reasoning.

Key challenges identified include handling intricate map visuals, spatial and temporal reasoning, and managing tasks requiring multi-step decision-making. This study highlights MAPEVAL’s potential to advance geo-spatial understanding in AI, suggesting future development could involve models specialized in spatial reasoning or enhanced integration of map tools.

**Strengths:**

1. Collecting relevant world information using map tools or APIs is a great way to test LLMs' ablility in solving geo-spatial reasoning with both internal and external information.
2. The inclusion of diverse task types of Place Information, Routing, Nearby, Unanswerable, and Trip effectively addresses a wide range of real-world user scenarios.
3. Great effort on using MapQaTor to streamline the collection of textual map data.
4. High quality human-labeled data and evaluation.
5. Broad distribution of data across various global regions.

**Weaknesses:**

1. Only multiple choice questions are adopted in MapEval. More diversity question type should be added to further explore the ability of LLMs, for real world problems are mostly open-ended.
2. The sample size for each task is limited, you may consider to use some methods to scale up your benchmark with great quality.
3. It would be more beneficial to test LLMs with different architectures, rather than focusing on different versions of the same model.

**Questions:**

1. There seem to be some problems with the API calling agent, can you explain more how you train the agent and possibly fix the problems of the agent?

---

> ### Author Response · Authors · 2024-11-23
> **Addressing W1**
>
> Thank you for your recognition of the applicability, effectiveness of our proposed benchmarking task, and comprehensive experiments in our work!
>
> > W1: Only multiple choice questions are adopted in MapEval. More diversity question type should be added to further explore the ability of LLMs, for real world problems are mostly open-ended.
>
> Thank you for your suggestion. In Section 3.1 (line 147 onwards) we discuss the motivations for MCQ based evaluation choice over open-ended ones. As discussed while open-ended are real-life but their accurate and reliable evaluation is challenging.  Therefore, we follow the widely adopted MCQ forms such as in MMLU [1], We note that while designing our queries, we deliberately choose MCQ for the evaluation purposes while keeping the queries flexible so that open-ended forms can also be made using our queries (simply without the options). Examples can be easily visible from Table 1, Table 12, and the qualitative examples in Appendix F.  Regarding diversity on more question types, please note that each of our question types can further be decompose into multiple sub-categories such as commonsense reasoning, distance computation, cardinal directional etc., (as discussed in line 243, Section 4.3, Section 5). However for readability we cluster them into Table 1 categories. However, we welcome suggestions for additional real-life map service scenarios from daily activities.
>
> **References:**
>
> [1] Hendrycks, Dan, et al. "Measuring massive multitask language understanding." arXiv preprint arXiv:2009.03300 (2020).

---

> ### Author Response · Authors · 2024-11-23
> **Addressing W2**
>
> > W2: The sample size for each task is limited, you may consider to use some methods to scale up your benchmark with great quality.
>
> **(Also discussed in Response to Reviewer DM4A W2-Part2 and W3)**
>
> Due to the high cost of both foundation models as well as tools/APIs, recent language agents often tend to evaluate on a small number of sub-sampled dataset. For example ReACT [1]  uses only 500 random samples from AlfWorld dataset, similarly Reflexion often tends to use [2] only 100 examples from HotpotQA). Therefore, recently proposed tool oriented or intense reasoning benchmark datasets are found to reasonable in size in order to be cost-effective: API-Bank: [3] (400 instances), Logical-reasoning benchmark LogiQA: [4] (641 examples), the most popular problem solving (code generation benchmarks) HumanEval: [5]  (164 instances only), CodeContests [6] (156 problems), Tao-bench [7] (165 problems), OS World [8] (369 problems), App world [9] (750 problems), TravelPlanner [10] (1.2K problems). Consequently, we carefully construct our problem instances balanced in size and covering different challenges.
>
> However, we thank you for the feedback and leveraging the rebuttal time-periods of ICLR, we strive to increase our problem set to 700 and ensure that each of the 54 countries is now represented by a minimum of 6 questions while maintaining the same quality control and pipeline. This improvement addresses the concern regarding insufficient representation of certain countries, making the evaluation more robust and geographically diverse. In our observation, further attempts in incorporating additional questions from different countries revealed no significant change in model performance. To maintain a cost-effective testbed, we opted not to include more instances, suggesting that our evaluation problem set already offers sufficient diversity for robust and reliable assessment. This improvement addresses the concern regarding insufficient representation of certain countries, making the evaluation more robust and geographically diverse.
>
> In addition, as outlined in Section 3.1, our goal is to benchmark on high standard real-user queries aiming to solve day-to-day actual Map application tasks and hence we do not scale up by synthesizing.
>
> **References:**
>
> [1] Yao, Shunyu, et al. "ReAct: Synergizing Reasoning and Acting in Language Models." The Eleventh International Conference on Learning Representations.\
> [2] Shinn, Noah, et al. "Reflexion: Language agents with verbal reinforcement learning." Advances in Neural Information Processing Systems 36 (2024).\
> [3] Li, Minghao, et al. "API-Bank: A Comprehensive Benchmark for Tool-Augmented LLMs." Proceedings of the 2023 Conference on Empirical Methods in Natural Language Processing. 2023.\
> [4] Liu, Jian, et al. "LogiQA: a challenge dataset for machine reading comprehension with logical reasoning." Proceedings of the Twenty-Ninth International Conference on International Joint Conferences on Artificial Intelligence. 2021.\
> [5] Chen, Mark, et al. "Evaluating large language models trained on code." arXiv preprint arXiv:2107.03374 (2021).\
> [6] Li, Yujia, et al. "Competition-level code generation with alphacode." Science 378.6624 (2022): 1092-1097.\
> [7] Yao, Shunyu, et al. "$\tau$-bench: A Benchmark for Tool-Agent-User Interaction in Real-World Domains." arXiv preprint arXiv:2406.12045 (2024).\
> [8] Xie, Tianbao, et al. "Osworld: Benchmarking multimodal agents for open-ended tasks in real computer environments." arXiv preprint arXiv:2404.07972 (2024).\
> [9] Trivedi, Harsh, et al. "AppWorld: A Controllable World of Apps and People for Benchmarking Interactive Coding Agents." Proceedings of the 62nd Annual Meeting of the Association for Computational Linguistics (Volume 1: Long Papers). 2024.\
> [10] Xie, Jian, et al. "TravelPlanner: A Benchmark for Real-World Planning with Language Agents". Poster presented at the Forty-First International Conference on Machine Learning (ICML), 2024.\

---

> ### Author Response · Authors · 2024-11-23
> **Addressing W3 and Q1**
>
> > W3: It would be more beneficial to test LLMs with different architectures, rather than focusing on different versions of the same model.
>
> We have made a concerted effort to include a comprehensive range of state-of-the-art (SOTA) foundation models that are related spanning numerous widely adopted model families/series such as GPTs, Geminis, Gemma, Claudes, Llamas, Mistrals, Qwens, Phi and so on (28 foundation models) as discussed in Section 4.1. However, as open-source models are often evaluated with their smaller versions due to the resource constraints, we also needed to evaluate these variations. During this rebuttal, as per the suggestions from other reviewers we also have conducted additional experiments using 7 different models such as the newest Qwen2.5. Therefore, we believe our evaluation is fair, comprehensive, not biased to any LLM family and already addresses this feedback. We would be happy to explore any particular model if you recommend one.
>
> > Q1: There seem to be some problems with the API calling agent, can you explain more how you train the agent and possibly fix the problems of the agent?
>
> It is unclear what you meant by “There seem to be some problems with the API calling agent”.  In our experimental protocol and setup (Section 4.1), we mention that for a fair evaluation of the foundation models for MapEval-API task we evaluate all models uniformly through the widely adopted tool usage agent ReAct. We adopt the Zero Shot ReAct framework implemented by langchain (https://python.langchain.com/v0.1/docs/modules/agents/agent_types/react/). We clarify that this is an inference-only method that simply prompts the corresponding LLMs under consideration  (e.g., instruct/chat versions of LLMs) for which no additional agent training is required.  We also discuss challenges in API calling agents such as infinite loops (in our error analysis in Section 4.3) or further strategies for building enhanced agents in Section 5 such as using Chameleon [1]. Please note that there is a huge scope in improving the API calling agent that is left as future learning approaches.
>
> **References:**
>
> [1] Lu, Pan, et al. "Chameleon: Plug-and-play compositional reasoning with large language models." Advances in Neural Information Processing Systems 36 (2024).

---

> ### Author Response · Authors · 2024-11-28
> **Reminder**
>
> Dear reviewer,
>
> It is a gentle reminder to acknowledge our rebuttal and make necessary reassessment of the score. Again, we thank you for your time and effort in reviewing our work.

---

### Official Review · Reviewer_DM4A · 2024-11-03

**Soundness:** 2
**Presentation:** 3
**Contribution:** 2
**Rating:** 5
**Confidence:** 4

**Summary:**

This paper explores the map-based understanding and reasoning abilities of large language models and vision language models. It introduces MapEval, a comprehensive benchmark designed to evaluate these foundation models using text, API, and vision environments, assessing their capabilities to handle heterogeneous geospatial contexts and perform compositional geo-spatial reasoning. MapEval comprises over 500 questions spanning 138 cities in 45 countries. Extensive experiments are conducted on three types of tasks within the MapEval framework, involving 25 advanced foundation models. The results highlight the impressive capabilities of state-of-the-art commercial foundation models, although they still lag behind human performance by more than 20%. Moreover, significant room for improvement remains for open-source large language models and vision language models.

**Strengths:**

1. The paper is well-structured and easy to follow.
2. The approach of assessing the geospatial reasoning abilities of advanced foundation models through map services is interesting. By leveraging real-world contexts and results provided by Google Maps, the evaluation becomes both practical and valuable.
3. The MapEval benchmark offers three distinct environments—text, API, and vision—allowing for a comprehensive assessment of foundation models. This setup provides flexibility to examine model performance from different perspectives while maintaining a consistent context.

**Weaknesses:**

1. The discussion of related work is insufficient. For example, the paper does not discuss the differences and relationships between this study and prior research, such as GeoQA [1] and other similar evaluations [2].

[1] Mai, Gengchen, et al. "Geographic question answering: challenges, uniqueness, classification, and future directions." AGILE: GIScience series 2 (2021): 8.

[2] Roberts, Jonathan, et al. "GPT4GEO: How a Language Model Sees the World's Geography." arXiv preprint arXiv:2306.00020 (2023).

2. The evaluation is relatively small in scale and lacks a systematic approach. The tasks outlined in Table 1 and Figure 2 are overly simplistic, requiring more expert-level geospatial reasoning knowledge to enhance and extend the design of the evaluation framework.
3. As shown in Figure 3, although the questions span 54 countries, many nations are represented by only a single question, which is insufficient to ensure a robust evaluation. The abstract's claim regarding spatial coverage seems exaggerated.
4. The set of large language models and vision-language models assessed in the experiments is limited. Some important models, such as the Qwen2.5 series, Qwen2VL series, VILA series, and MiniCPM2.5, should be included to provide a more comprehensive evaluation.

**Questions:**

1. Could the authors elaborate on the relationship between MapEval and prior research, e.g.,  GeoQA and GPT4GEO?
2. The paper should report additional results for significant foundation models, such as the Qwen2.5 series, Qwen2VL series, VILA series, and MiniCPM2.5.

---

> ### Author Response · Authors · 2024-11-23
> **Addressing W1, Q1**
>
> Thank you for your recognition of the applicability, effectiveness of our proposed benchmarking task, and comprehensive experiments in our work!
> >W1, Q1: The discussion of related work is insufficient. For example, the paper does not discuss the differences and relationships between this study and prior research, such as GeoQA and GPT4GEO.
>
> We would like to clarify that GeoQA [12] was indeed discussed in our paper, specifically in Section 2. However, we acknowledge that the discussion may have been brief and could have benefited from additional detail. Based on your feedback, we now provide a more detailed comparison.
>
> **Comparison with GeoQA Approaches:**
>
> Template-based GeoQA models ([2-6]) have predominantly followed a two-step strategy for answering geographic questions: (1) given a natural language query, selecting a template form (e.g., sql template) from a predefined list and (2) using these templates to query structured geographic knowledge sources (e.g., DBpedia [9], YAGO [11], Freebase [10], and OpenStreetMap). Overall, these approaches focus primarily on converting natural language questions into structured query language (sql) scripts. For instance, GeoQuestions1089 [6] contains 1,089 questions paired with their corresponding templated database queries (GeoSPARQL [7]) and answers derived from the geospatial knowledge graph YAGO2geo [8].
>
> Our MapEval framework takes a fundamentally different approach. In MapEval-Textual, we shift from database querying or sql generation to evaluating geospatial reasoning in Large Language Models (LLMs). Annotators collect factual map services data (e.g., from GoogleMap), which serves as context for LLMs, allowing us to isolate and evaluate their reasoning capabilities rather than their knowledge retrieval. MapEval-API adopts a more practical approach by enabling direct interaction with map APIs to answer questions. Thus, in MapEval, LLMs are responsible for answering questions directly, unlike previous works where models generated database queries (e.g., Geoquery, GeoSPARQL) to retrieve answers from external knowledge bases. In addition, MapEval-vision employs multi-modal challenges.
>
> **Comparison with GPT4GEO:**
>
> GPT4GEO [1] examined GPT-4's geographic knowledge by probing it through template-based queries without API (plugins) or Internet access. Their evaluation focused on analyzing a single model using templated queries about generic location and direction-oriented facts, such as routing, navigation, and planning for well-known cities and places (e.g., renowned cities, shipping ports, airports, and coarse travel times like "8-hour flights to west coast" or  plans like Frankfurt to Brussels: From Frankfurt, you'll need to take a train to Brussels, Belgium. This is a 3-hour train ride). While the findings suggest that GPT-4 shows promising geo-spatial knowledge, this approach neither establishes a benchmark for geo-spatial reasoning nor incorporates real-life user queries or map services (e.g., Google Maps) as a geospatial information base.
>
> Our approach employs fundamentally different evaluation and design principles (outlined in Section 3.1). We establish a benchmarking of deeper geo-spatio-temporal reasoning capabilities across multiple foundation models using real user queries rather than templates. Uniquely, our evaluation encompasses multimodal understanding, tool interactions, and answerability determination. Additionally, we provide foundation models with fine-grained map services data through both context and API access, enabling a more comprehensive benchmarking of their geospatial question-answering abilities.
>
>
> For more in-depth details, a comprehensive related work discussion has also been added in the Appendix A of the revised paper. Additionally, the related work section in the main text has been updated to address the feedback.
>
> **References:**
>
> [1] Roberts, J., et al. "GPT4GEO: How a Language Model Sees the World's Geography." arXiv, 2023.\
> [2] Zelle, J. M., and Mooney, R. J. "Learning to parse database queries using inductive logic programming." AAAI, 1996.\
> [3] Chen, W., et al. "A synergistic framework for geographic QA." IEEE ICSC, 2013.\
> [4] Chen, W. "Parameterized Spatial SQL Translation for GeoQA." IEEE ICSC, 2014.\
> [5] Punjani, D., et al. "Template-based QA over linked geospatial data." GIR, 2018.\
> [6] Kefalidis, S.-A., et al. "Benchmarking GeoQA using GeoQuestions1089." ISWC, 2023.\
> [7] OGC. "GeoSPARQL: A Geographic Query Language for RDF Data." Open Geospatial Consortium, 2011.\
> [8] Karalis, N., et al. "Extending YAGO2 with geospatial knowledge." ISWC, 2019.\
> [9] Auer, S., et al. "Dbpedia: A nucleus for a web of open data." ISWC, 2007.\
> [10] Bollacker, K., et al. "Freebase: A shared database of structured knowledge." AAAI, 2007.\
> [11] Suchanek, F. M., et al. "Yago: a core of semantic knowledge." WWW, 2007.\
> [12] Mai, G., et al. "Geographic question answering: Challenges and future directions." AGILE GIScience, 2021.

---

> ### Author Response · Authors · 2024-11-23
> **Addressing W2-part-1**
>
> > W2-part-1: The evaluation is relatively small in scale and lacks a systematic approach.
>
> Due to the high cost of both foundation models as well as tools/APIs, recent language agents often tend to evaluate on a small number of sub-sampled dataset. For example ReACT [1]  uses only 500 random samples from AlfWorld dataset, similarly Reflexion often tends to use [2] only 100 examples from HotpotQA. Therefore, recently proposed tool oriented or intense reasoning benchmark datasets are found to be reasonable in size in order to be cost-effective: API-Bank: [3]  (400 instances), Logical-reasoning benchmark LogiQA: [4] (641 examples), the most popular problem solving (code generation benchmarks) HumanEval: [5]  (164 instances only), CodeContests [6] (156 problems), Tau-bench [7] (165 problems), OS World [8] (369 problems), App world [9] (750 problems), TravelPlanner [10] (1.2K problems). Consequently, we carefully construct our problem instances balanced in size and covering different challenges.
>
> However, we thank you for the feedback and leveraging the rebuttal time-periods of ICLR, we strive to increase our problem set to 700 while maintaining the same high quality. In addition, we believe that our contributions in evaluation strategies (such as three tasks with different challenges, utilizing multiple-choice question (MCQ) setups that enable reliable accuracy-based assessments, while also allowing for expansion into open-ended formats, evaluating all foundation models uniformly with the same ReACT agent for API task, etc), detailed performance studies, qualitative examples, fine-grained error analysis and newly added strategies for improvements, together constitute a complete benchmarking study.
>
> All the statistics, results, error analysis, and strategies for model improvements are updated in the paper (in blue color).
> In case you have any additional evaluation criteria in mind to further enhance MapEval, please feel free to specify so that we can address it.
>
> **References:**
>
> [1] Yao, Shunyu, et al. "ReAct: Synergizing Reasoning and Acting in Language Models." The Eleventh International Conference on Learning Representations.\
> [2] Shinn, Noah, et al. "Reflexion: Language agents with verbal reinforcement learning." Advances in Neural Information Processing Systems 36 (2024).\
> [3] Li, Minghao, et al. "API-Bank: A Comprehensive Benchmark for Tool-Augmented LLMs." Proceedings of the 2023 Conference on Empirical Methods in Natural Language Processing. 2023.\
> [4] Liu, Jian, et al. "LogiQA: a challenge dataset for machine reading comprehension with logical reasoning." Proceedings of the Twenty-Ninth International Conference on International Joint Conferences on Artificial Intelligence. 2021.\
> [5] Chen, Mark, et al. "Evaluating large language models trained on code." arXiv preprint arXiv:2107.03374 (2021).\
> [6] Li, Yujia, et al. "Competition-level code generation with alphacode." Science 378.6624 (2022): 1092-1097.\
> [7] Yao, Shunyu, et al. "$\tau$-bench: A Benchmark for Tool-Agent-User Interaction in Real-World Domains." arXiv preprint arXiv:2406.12045 (2024).\
> [8] Xie, Tianbao, et al. "Osworld: Benchmarking multimodal agents for open-ended tasks in real computer environments." arXiv preprint arXiv:2404.07972 (2024).\
> [9] Trivedi, Harsh, et al. "AppWorld: A Controllable World of Apps and People for Benchmarking Interactive Coding Agents." Proceedings of the 62nd Annual Meeting of the Association for Computational Linguistics (Volume 1: Long Papers). 2024.\
> [10] Xie, Jian, et al. "TravelPlanner: A Benchmark for Real-World Planning with Language Agents". Poster presented at the Forty-First International Conference on Machine Learning (ICML), 2024.

---

> ### Author Response · Authors · 2024-11-23
> **Addressing W2-part-2**
>
> > W2-part-2:  The tasks outlined in Table 1 and Figure 2 are overly simplistic, requiring more expert-level geospatial reasoning knowledge to enhance and extend the design of the evaluation framework.
>
> We thank you for giving this feedback which enables us to further clarify the scope and complexity of our tasks and bring in a new set of analyses. While designing MapEval, our goal was not to synthesize linguistically complex questions, rather we ensure the data properties outlined in Section 3.1 such as real-user queries aiming to solve day-to-day actual Map application tasks. We argue that while some of these real-life queries (e.g., in PlaceInfo, and Routing category in Textual/API tasks) may appear simplistic, they often require deeper spatio-temporal reasoning to answer and domain know-how (i.e., beyond memorization). As mentioned in line 30, our benchmarking reveals that all models still fall short of human performance significantly (e.g., by more than 20% on average). In section 4.3, we identify a number of reasoning challenges in our tasks and refer to the full details of examples (both queries, contexts, options, and model outputs) in Appendix F listings to understand fully. In addition, we updated Table 1 (in the main paper), and presented Table 12 in Appendix to show additional complex queries. However, for your convenience we highlight some examples below.
>
> - What is the direction of the Seurasaari Open-Air Museum from Helsinki Central Station? Options: East, West, North, South (Listing 2, Page 23, Appendix F).
>
> Although it seems a simple question but to answer this question, the model (i) needs to identify the latitude and longitude (co-ordinates) of both Seurasaari and Helsinki Central Station (ii) Resolve two directions in terms of both latitude and longitude from the co-ordinates (iii) Determine the single major direction from them and respond.
>
> Similar other linguistically simple yet challenging examples include:
>
> - In the Nearby category, there are more intricate queries, such as those that incorporate the buffer concept to determine Points of Interests (POIs) within a specified distance of a given location. Example of some complex questions: “How many shopping malls are there within a 500 m radius of Berlin Cathedral?” “I am at Toronto Zoo. Today is Sunday and it's currently 8:30 PM. How many nearby ATMs are open now?”
>
> - In the Routing category, finding the desired route from the step-by-step navigation is challenging due to long context and calculation. Example of some complex questions: “I want to walk from D03 Flame Tree Ridge to Aster Cedars Hospital, Jebel Ali. Which walking route involves taking the pedestrian overpass?” “On the driving route from Hassan II Mosque to Koutoubia via A3, how many roundabouts I will encounter in total?”

---

> ### Author Response · Authors · 2024-11-23
> **Addressing W3, W4 and Q2**
>
> > W3: As shown in Figure 3, although the questions span 54 countries, many nations are represented by only a single question, which is insufficient to ensure a robust evaluation.
>
> We appreciate the reviewer's observation regarding the spatial coverage of our dataset. In response, leveraging the rebuttal period, we have further annotated more data and increased the dataset size to 700 and ensured that each of the 54 countries is now represented by a minimum of 6 questions (see updated table 13) while maintaining the same quality control and pipeline. This improvement addresses the concern regarding insufficient representation of certain countries, making the evaluation more robust and geographically diverse. In our observation, further attempts in incorporating additional questions from different countries revealed no significant change in model performance. To maintain a cost-effective testbed, we opted not to include more instances, suggesting that our evaluation problem set already offers sufficient diversity for robust and reliable assessment.
>
> > W4 and Q2: The set of large language models and vision-language models assessed in the experiments is limited. Some important models, such as the Qwen2.5 series, Qwen2VL series, VILA series, and MiniCPM2.5, should be included to provide a more comprehensive evaluation.
>
> We have evaluated Qwen-2.5 (7B, 14B, 72B), Qwen2-VL-7B, MiniCPM-Llama3-V-2_5 and Llama-3-VILA1.5-8B. All the results and analyses have been updated in the paper. Thanks for your valuable suggestion, as Qwen2-VL becomes the best performing open source VLM and Qwen-2.5-72B becomes the top open source LLM in the hardest textual task ‘Trip’.

---

> ### Author Response · Authors · 2024-11-28
> **Reminder**
>
> Dear reviewer,
>
> It is a gentle reminder to acknowledge our rebuttal and make necessary reassessment of the score. Again, we thank you for your time and effort in reviewing our work.

---

> > ### Comment · Reviewer_DM4A · 2024-12-01
> >
> > Thank you very much for the author's detailed description and the additional experimental results. These have addressed some of my concerns, I will maintain my original score.

---

> ### Author Response · Authors · 2024-12-01
> **Response to Reviewer’s Feedback and Further Clarification Request**
>
> Dear Reviewer,
>
> Thank you very much for taking the time to give our paper a second look and for reviewing our responses and the additional experimental results. We greatly appreciate your thoughtful consideration and for acknowledging that we have addressed some of your concerns.
>
> However, it is possible that some points may need more discussion. We would be grateful if you could kindly clarify the concerns that remain, so that we can continue to refine and improve our work.
>
> Thank you once again for your valuable feedback.

---

### Author Response · Authors · 2024-11-26

Dear Reviewers,

As we approach the end of the rebuttal phase, with just one day remaining for PDF revisions, we kindly ask if you could take a second look at our paper and the authors’ responses. Please let us know if you have any remaining concerns.

---

### Author Response · Authors · 2024-11-30

### **Dear Reviewers, ACs, and PCs,**

We sincerely thank all the reviewers for their valuable and constructive feedback, as well as for dedicating their time to reviewing our paper. Based on the insightful suggestions provided, we have conducted a thorough revision to address the reviewers' key concerns. Below, we summarize these concerns and outline the revisions and updates included in the final submission. We hope this clarifies the progress and outcomes of our rebuttal for the reviewers, ACs, and PCs.

---

### **Concerns and Revisions**

- **[DM4A, bA3F, WpVB]** The number of test cases available for evaluating foundation models is relatively small and  many nations are represented by only a single question.
  - **[Authors]** Increased the dataset size from 573 to 700 with at least 6 questions per country, while maintaining the same quality control and pipeline. Incorporating additional questions from different countries revealed no significant change in models' performance. To maintain a cost-effective testbed, we opted not to include further additional instances.
- **[DM4A, bA3F]** The paper should report additional results for significant foundation models, such as the Qwen2.5 series, Qwen2VL series, VILA series, and MiniCPM2.5.
  - **[Authors]** Added evaluation of Qwen-2.5 (7B, 14B, 72B), Gemma-2.0-27B, Qwen2-VL-7B, MiniCPM-Llama3-V-2_5 and Llama-3-VILA1.5-8B.
- **[bA3F, rmq7, WpVB]** Paper might be lacking in terms of analyzing the distinct capabilities demonstrated by different models based on the evaluation results.
  -  **[Authors]** Added numerical analyses of LLMs’ performance (*MapEval-Textual*) in three new subcategories (Straight-Line Distance, Cardinal Direction, Counting) to explore the strengths and weaknesses.
- **[rmq7]** How to design a model that performs well across all aspects?
  - **[Authors]** Integrated external tools (e.g., a calculator) into the LLM to compute straight-line distance and cardinal direction. This integration resulted in a significant performance improvement in *MapEval-Textual*.
- **[bA3F, rmq7]** How to possibly fix the problems of the API-calling agent?
  - **[Authors]** Adapted Chameleon (a multi-agent system) Framework to the *MapEval-API*, which resulted in noticeable improvement in performance.
- **[DM4A]** The discussion of related work is insufficient.
  - **[Authors]** Added detailed related work section in Appendix A. Additionally, the related work section in the main text has been updated.
- **[DM4A]** The tasks outlined in Table 1 and Figure 2 are overly simplistic.
  -  **[Authors]** Updated Table 1 and added additional list of complex question examples in Appendix F. We argue that while some of these real-life queries may appear simplistic, they often require deeper spatio-temporal reasoning to answer. As acknowledged by **bA3F**, these tasks effectively addresses a wide range of real-world user scenarios.
- **[WpVB]** Why were geographic specialized foundation models (e.g., K2) not used?
  - **[Authors]** We evaluated K2 (which is designed specifically for geoscience-related tasks), but it achieved only 20.33% accuracy in *MapEval-Textual*. Besides, at the time of our evaluation, there were no geospatial specialized foundation models that met the specific requirements of our task.
- **[WpVB]** Why were remote sensing models not evaluated in *MapEval-Visual*?
  - **[Authors]** Our study does not involve remote sensing images. Instead, we provide digital map view snapshots to VLMs. A more detailed comparison is added in Appendix A.3.
---

We sincerely thank the reviewers for their valuable suggestions, which have helped us strengthen our revised submission.

*Best regards,*

*Authors*

---

> ### Author Response · Authors · 2024-12-03
>
> **Dear Reviewers**,
>
> We sincerely thank you for your hard work during the review process. We have carefully read your comments, provided additional clarifications, and made revisions to the paper. We are more than happy to engage in discussions with you, as it plays a crucial role in resolving your concerns and improving our paper!
>
> **However, as the rebuttal period will conclude in two days, we would be grateful if you could let us know whether we have addressed your concerns at your earliest convenience. If you still have any concerns, we would be delighted to address them promptly. If your concerns have been addressed, we would appreciate it if you could kindly reassess our work and increase the rating.**

---

### Meta-Review · Area_Chair_ZQRS · 2024-12-12

**Metareview:**

The paper introduces MAPEVAL, a benchmark designed to evaluate the geo-spatial reasoning capabilities of foundation models across textual, API-based, and visual tasks. MAPEVAL encompasses 700 diverse multiple-choice questions derived from real-world map interactions across 180 cities in 54 countries, addressing challenges in handling spatial relationships, navigation, and map infographics. Evaluations of 28 foundation models reveal that while top performers like Claude-3.5-Sonnet, GPT-4o, and Gemini-1.5-Pro show promising results, they lag behind human performance by over 20%. The benchmark highlights significant gaps in complex tasks like multi-step trip planning and rigorous map image reasoning, emphasizing MAPEVAL’s role in advancing AI’s geo-spatial understanding and real-world utility. The dataset and evaluation codes are open-sourced to foster research in this domain.

Overall I think the paper does make contributions to the community with a dataset and benchmark. The paper is well-structured and easy to follow. The approach of assessing the geospatial reasoning abilities of advanced foundation models through map services is interesting. By leveraging real-world contexts and results provided by Google Maps, the evaluation becomes both practical and valuable.

However, all reviewers have some issues, e.g., insufficient data scale and model evaluations. Some of the concerns are addressed and the paper is significantly improved. However, the novelty of this paper is around borderline so all the reviewers do not increase the score. The problem has been studied and the new datasets are not really large-scale. As this is a dataset/benchmarking paper, I believe it is very important to position itself and has more audience. As one of the reviewers point out, it is better to discuss why different models perform differently on the collected dataset, and exploring the relationship between the existing results and model designs to provide more insights.

**Additional Comments On Reviewer Discussion:**

I think the main disadvantages of the original versions by reviewers are summarized as follows:
- The evaluation is relatively small in scale and lacks a systematic approach.
- The set of large language models and vision-language models assessed in the experiments is limited. Some important models, such as the Qwen2.5 series, Qwen2VL series, VILA series, and MiniCPM2.5, should be included to provide a more comprehensive evaluation.

The authors truly do more experiments on more tests (up to 700) and evaluate several new models on the benchmark. However, the reviewers are not fully convinced. As this is a dataset/benchmarking paper, I believe it is very important to position itself and has more audience. As one of the reviewers point out, it is better to discuss why different models perform differently on the collected dataset, and exploring the relationship between the existing results and model designs to provide more insights.

---

### Decision · Program_Chairs · 2025-01-22

Reject